# SDDBench: A Benchmark for Synthesizable Drug Design

## Abstract

A significant challenge in wet lab experiments with current drug design generative models is the trade-off between pharmacological properties and synthesizability. Molecules predicted to have highly desirable properties are often difficult to synthesize, while those that are easily synthesizable tend to exhibit less favorable properties. As a result, evaluating the synthesizability of molecules in general drug design scenarios remains a significant challenge in the field of drug discovery. The commonly used synthetic accessibility (SA) score aims to evaluate the ease of synthesizing generated molecules, but it falls short of guaranteeing that synthetic routes can actually be found. Inspired by recent advances in top-down synthetic route generation and forward reaction prediction, we propose a new, data-driven metric to evaluate molecule synthesizability. This novel metric leverages the synergistic duality between retrosynthetic planners and reaction predictors, both of which are trained on extensive reaction datasets. To demonstrate the efficacy of our metric, we conduct a comprehensive evaluation of round-trip scores across a range of representative molecule generative models.

## 1 Introduction

Drug design is a fundamental problem in machine learning for drug discovery. However, when these computationally predicted molecules are put to the test in wet lab experiments, a critical issue often arises: many of them prove to be unsynthesizable in practice (Parrot et al., 2023). This synthesis gap can be attributed to two primary factors. Firstly, while structurally feasible, the predicted molecules often lie far beyond the known synthetically-accessible chemical space (Ertl & Schuffenhauer, 2009). This significant departure from known chemical territory makes it extremely difficult, and often impossible, to discover feasible synthetic routes (Segler et al., 2018; Liu et al., 2023b). This synthesis challenge is underscored by numerous clinical drugs derived from natural products, which, due to their intricate structures, can only be obtained through direct extraction from natural sources rather than synthesis methods (Zheng et al., 2022). These natural products often have complex ring structures and multiple chiral centers, which makes their chemical synthesis challenging (Paterson & Anderson, 2005). Additionally, the biological processes that create these compounds are frequently not well understood, increasing the complexity of laboratory synthesis. Secondly, even when plausible reactions are identified based on literature, they may fail in practice due to the inherent complexity of chemistry (Lipinski, 2004). The sensitivity of chemical reactions is such that even minor changes in functional groups can potentially prevent a reaction from happening as anticipated.

The ability to synthesize designed molecules on a large scale is crucial for drug development. Some current methods (You et al., 2018; Gao & Coley, 2020) rely on the Synthetic Accessibility (SA) score (Ertl & Schuffenhauer, 2009) for synthesizability evaluation. This score assesses how easily a drug can be synthesized by combining fragment contributions with a complexity penalty. However, this metric has limitations as it evaluates synthesizability based on structural features and fails to account for the practical challenges involved in developing actual synthetic routes for these molecules. In other words, a high SA score does not guarantee that a feasible synthetic route for the molecule can be identified using available molecule synthesis tools (Genheden et al., 2020; Tripp et al., 2022).

To overcome the limitations of the SA score, recent works (Cretu et al., 2024) have employed retrosynthetic planners or AiZynthFinder (Genheden et al., 2020) to evaluate the synthesizability of generated molecules. These tools are used to find synthetic routes and assess the proportion of

molecules for which routes can be found. As a result, these works rely on the search success rate for evaluating molecule synthesizability. However, this metric is overly lenient, as it fails to ensure that the proposed routes are actually capable of synthesizing the target molecules (Liu et al., 2023b). In practice, many reactions predicted by these tools may not be simulated in the wet lab, as these tools often rely on data-driven retrosynthesis models prone to predicting unrealistic or hallucinated reactions Zhong et al. (2023); Tripp et al. (2024).

To address the overly lenient evaluation metrics in previous retrosynthesis studies, where success is often defined merely by finding a "solution" without any regard to whether the solution can be executed in the wet lab (Tripp et al., 2024), FusionRetro (Liu et al., 2023b) proposes assessing whether the starting materials[1] of a predicted route of a target molecule match those in reference routes from the literature database for a target molecule. However, for new molecules generated by drug design models, reference synthetic routes are often unavailable in literature databases. This raises a critical question:

*Can data-driven retrosynthetic planners be used to evaluate the synthesizability of these molecules?*

Inspired by recent advancements that leverage forward reaction models (Sun et al., 2021) to enhance retrosynthesis algorithms and rank the top-k synthetic routes predicted by retrosynthetic planners (Schwaller et al., 2019b; Liu et al., 2024a), we propose a three-stage approach that incorporates forward reaction models for evaluating molecule synthesizability to address this question.

Our evaluation process consists of three stages. In the first stage, we use a retrosynthetic planner to predict synthetic routes for molecules generated by drug design generative models. In the second stage, we assess the feasibility of these routes using a reaction prediction model as a simulation agent, serving as a substitute for wet lab experiments. This model attempts to reconstruct both the synthetic route and the generated molecule, starting from the predicted route's starting materials. In the third stage, we calculate the Tanimoto similarity, also called the round-trip score, between the reproduced molecule and the originally generated molecule as the synthesizability evaluation metric. Our proposed metric also draws inspiration from evaluation methods used in image generation, such as the CLIP score (Radford et al., 2021; Hessel et al., 2021). In image generation, the CLIP score assesses the similarity between generated images and their corresponding text descriptions using the pre-trained CLIP model (Radford et al., 2021). Analogously, our point-wise round-trip score evaluates whether the starting materials in a predicted synthetic route can successfully undergo a series of reactions to produce the generated molecule.

With the round-trip score as the foundation, we develop a new benchmark to evaluate the "synthesizability" of molecules predicted by current structure-based drug design (SBDD) generative models. Our contributions can be summarized as follows:

- We recognize the limitations of the current metrics used for evaluating molecule synthesizability. Therefore, we propose the round-trip score as a metric to evaluate the synthesizability of new molecules generated by drug design models.
- We develop a new benchmark based on the round-trip score to evaluate existing generative models' ability to predict synthesizable drugs. This benchmark aims to shift the focus of the entire research community towards synthesizable drug design.

## 2 BACKGROUND

In this section, we discuss the details of drug design and molecule synthesis. Machine learning algorithms for molecule synthesis can be categorized into three main types: forward reaction prediction models, backward retrosynthesis prediction models, and search algorithms.

### 2.1 STRUCTURE-BASED DRUG DESIGN

While our newly developed benchmark is capable of evaluating a wide range of drug design models, this work specifically focuses on assessing the synthesizability of molecules generated by

---

[1]Starting materials are defined as commercially purchasable molecules. ZINC (Sterling & Irwin, 2015) provides open-source databases of purchasable compounds, and we define the compounds listed in these databases as our starting materials.

SBDD models. The primary goal of SBDD is to generate ligand molecules capable of binding to a specific protein binding site. In this context, we represent the target protein and ligand molecule as $\boldsymbol{p} = \{(\boldsymbol{x}_i^{\boldsymbol{p}}, \boldsymbol{v}_i^{\boldsymbol{p}})\}_{i=1}^{N_p}$ and $\boldsymbol{m} = \{(\boldsymbol{x}_i^{\boldsymbol{m}}, \boldsymbol{v}_i^{\boldsymbol{m}})\}_{i=1}^{N_m}$, respectively. Here, $N_p$ and $N_m$ denote the number of atoms in the protein $\boldsymbol{p}$ and ligand $\boldsymbol{m}$. For each atom, $\boldsymbol{x} \in \mathbb{R}^3$ represents its position in three-dimensional space, while $\boldsymbol{v} \in \mathbb{R}^K$ encodes its type. The core challenge of SBDD lies in accurately modeling the conditional distribution $P(\boldsymbol{m} \mid \boldsymbol{p})$.

## 2.2 REACTION PREDICTION

Reaction prediction aims to determine the outcome of a chemical reaction. The task involves predicting the products $\boldsymbol{\mathcal{M}}_p = \{\boldsymbol{m}_p^{(i)}\}_{i=1}^n \subseteq \boldsymbol{\mathcal{M}}$ given a set of reactants $\boldsymbol{\mathcal{M}}_r = \{\boldsymbol{m}_r^{(i)}\}_{i=1}^m \subseteq \boldsymbol{\mathcal{M}}$, where $\boldsymbol{\mathcal{M}}$ represents the space of all possible molecules. It's worth noting that in current public reaction datasets, such as USPTO (Lowe, 2014), only the main product is typically recorded (i.e., $n = 1$), with by-products often omitted. This simplification, while practical for many applications, has a limitation in capturing the full complexity of chemical reactions.

Figure 1: For a given molecule, multiple synthetic routes can be identified within the reaction database, illustrating the diverse routes available for its synthesis.

## 2.3 RETROSYNTHESIS PREDICTION

Retrosynthesis, the inverse process of reaction prediction, aims to identify a set of reactants $\boldsymbol{\mathcal{M}}_r = \{\boldsymbol{m}_r^{(i)}\}_{i=1}^m \subseteq \boldsymbol{\mathcal{M}}$ capable of synthesizing a given product molecule $\boldsymbol{m}_p$ through a single chemical reaction. This process essentially works backward from the desired product, determining the precursor molecules necessary for its synthesis. By doing so, retrosynthesis plays a crucial role in planning synthetic routes for complex molecules, particularly in drug discovery and materials science.

## 2.4 REACTION PREDICTION (FORWARD) VS. RETROSYNTHESIS PREDICTION (BACKWARD)

Reaction prediction and retrosynthesis prediction differ fundamentally in their nature and objectives. Reaction prediction is a deterministic task, where specific reactants under given conditions typically yield a predictable outcome. In contrast, retrosynthesis prediction is inherently a one-to-many task, providing multiple potential routes to a desired product as illustrated in Figure 1.

## 2.5 RETROSYNTHETIC PLANNING

Retrosynthetic planning is a strategic approach to predict synthetic routes for target molecules. This process works backward from the desired target, identifying potential precursor molecules that could be transformed into the target through chemical reactions. These precursors are then further decomposed into simpler, readily available starting materials or building blocks. A synthetic route can be formally represented as a tuple with four elements: $\boldsymbol{\mathcal{T}} = (\boldsymbol{m}_{tar}, \boldsymbol{\tau}, \boldsymbol{\mathcal{I}}, \boldsymbol{\mathcal{B}})$, where $\boldsymbol{m}_{tar} \in \boldsymbol{\mathcal{M}} \backslash \boldsymbol{\mathcal{S}}$ is the target molecule, $\boldsymbol{\mathcal{S}} \subseteq \boldsymbol{\mathcal{M}}$ represents the space of starting materials, $\boldsymbol{\mathcal{B}} \subseteq \boldsymbol{\mathcal{S}}$ denotes the specific starting materials used, $\boldsymbol{\tau}$ is the series of reactions leading to $\boldsymbol{m}_{tar}$, and $\boldsymbol{\mathcal{I}} \subseteq \boldsymbol{\mathcal{M}} \backslash \boldsymbol{\mathcal{S}}$ represents the intermediates. In organic synthesis, a "route" refers to the complete flowchart of reactions required to synthesize a target molecule, as illustrated in Figure 1. This definition differs from its usage in computer science. Synthetic routes can be classified as convergent (Figure 1) or non-convergent, depending on whether the reactions within the route have branching points (Gao et al., 2022a). The planning process is iterative. At each step, single-step retrosynthesis models predict various sets of potential reactants that could lead to the desired product. A search algorithm then selects the most promising solutions to extend the synthetic route further. This process continues until all leaf nodes correspond to readily available starting materials, resulting in a complete synthetic route from purchasable molecules to the target compound.

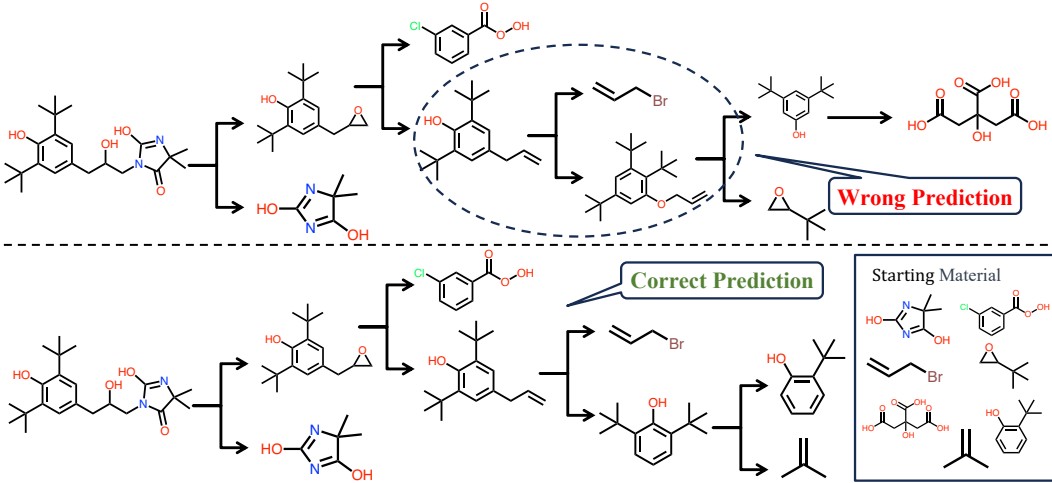

Figure 2: Comparison of evaluation metrics for retrosynthetic planning. The search success rate deems both routes successful, while the matching-based metric correctly identifies the top route as incorrect and the bottom route as correct, demonstrating its superior reliability.

## 2.6 EVALUATION OF MOLECULE SYNTHESIS

Current evaluation methods for single-step reaction and retrosynthesis predictions primarily rely on the exact match metric. This approach assesses whether the predicted results match the ground truth in the test dataset. Typically, multiple predictions are generated, and the top-k test accuracy is reported.

Until recently, evaluation criteria for retrosynthetic planning had not reached a clear consensus, but they have now converged on a few key metrics. Among these, one of the most widely used is the success rate of finding a synthetic route within a limited number of calls for single-step retrosynthesis prediction (typically capped at 500). However, this metric, known as the search success rate, is overly lenient as it does not verify whether the searched synthetic route can be executed in the wet lab to synthesize the target molecule. This limitation is particularly problematic for targets requiring long synthetic routes, where errors can accumulate across multiple steps.

To illustrate this limitation, we observe that existing single-step models achieve top-5 accuracies of less than 80% (Somnath et al., 2021). In contrast, current retrosynthetic planning methods report search success rates exceeding 99% (Xie et al., 2022) under the limit of 500 single-step retrosynthesis prediction iterations. This discrepancy is counterintuitive since longer synthetic routes should inherently have a lower likelihood of success due to the increasing complexity of the synthesis. This raises concerns about the quality of the routes deemed "successful" by multi-step planners.

To address the limitations of using search success rate as an evaluation metric for retrosynthetic planning, FusionRetro (Liu et al., 2023b) introduces a matching-based evaluation approach. This method compares the starting materials of synthetic routes predicted by retrosynthetic planners for target molecules with those from reference routes retrieved from literature databases. If the starting materials of a predicted route match those of any reference route, the prediction is considered accurate and successful. This approach aligns with the evaluation methodology used in single-step retrosynthesis, which also relies on literature databases. Moreover, FusionRetro goes further by constructing a reaction network from all reactions in the literature database. For a given target molecule, it extracts all synthetic routes from this reaction network, with the leaf nodes within these routes being the starting materials. As FusionRetro reports, this often enables the identification of multiple synthetic routes for a target molecule within the literature database.

While this matching-based approach has its limitations such as the inability of current literature databases to cover all equivalent synthetic routes to a target molecule, this limitation is not unique to FusionRetro and also applies to existing retrosynthesis evaluation methods. Despite this, the

Figure 3: Illustration of the round-trip score calculation process. It consists three stages: Retrosynthetic Planning, Forward Reproduction, and Similarity Computation.

matching-based evaluation metric is a more reliable and rigorous alternative to the search success rate.

To illustrate the differences between the two metrics, we introduce an example in Figure 2. The top part of the figure shows a predicted synthetic route where the retrosynthesis prediction within the highlighted circle is incorrect, while the bottom part illustrates a correctly predicted synthetic route. Despite this, both routes are considered successful under the search success rate metric. This is because the search success rate evaluates the solvability of finding a route with leaf nodes as starting materials within a limited number of single-step retrosynthesis prediction iterations for a given target molecule.

In contrast, the starting material matching-based metric clearly distinguishes between the two routes. It identifies the top route as incorrect and the bottom route as correct, as the incorrect route in the top example would not match any entries in the literature reaction database. This example intuitively demonstrates that the matching-based evaluation metric provides a more reliable and accurate assessment than the search success rate.

## 3 ROUND-TRIP SCORE

In this section, we introduce a novel metric called the round-trip score. This metric is designed to assess the feasibility of synthetic routes for molecules generated by drug design models. Specifically, it evaluates the probability that retrosynthetic planners, trained on current reaction data, can successfully predict feasible synthetic routes for these proposed molecules.

### 3.1 MOTIVATION

As discussed in Sections 1 and 2, current heuristics-based metrics for evaluating molecule synthesizability, such as the SA score, fail to ensure that synthetic routes can be identified using existing data-driven molecule synthesis tools. However, these tools typically rely on the search success rate as a metric, which does not assess the feasibility of the predicted routes. We have identified a critical flaw in the search success rate metric in Section 2.6 and find that the match-based evaluation metric provides a more reliable alternative.

However, for new molecules generated by drug design generative models, reference routes are often missing from literature databases, making it impossible to evaluate predicted routes using match-based metrics. Ideally, the most accurate evaluation would involve directly validating the predicted

routes in the wet lab to confirm whether they can synthesize the target molecules. However, this approach is prohibitively expensive, especially when evaluating large numbers of molecules.

To address this challenge, we note that recent forward reaction prediction models achieve top-1 accuracies exceeding 90% (Bi et al., 2021). These models can simulate reactions to verify whether the predicted synthetic routes are capable of synthesizing the target molecules. While this approach has its limitations, it is far more reliable than the search success rate and avoids the significant costs associated with wet lab experiments.

### 3.2 THREE-STAGE EVALUATION PROCESS

In this section, we discuss the details of three stages of our evaluation method: retrosynthetic planning, forward reaction prediction, and similarity computation.

Given a molecule $m$ proposed by a generative model, we first use a retrosynthetic planner to predict a synthetic route. Starting from the initial materials of this route, we then employ a reaction model to simulate wet lab experiments and reproduce the synthetic route until we reach the final molecule $m'$. Finally, we compute the Tanimoto similarity between $m$ and $m'$, which we define as the round-trip score. Figure 3 provides an illustration of the entire process. The round-trip score, which encapsulates this process, can be mathematically expressed as follows:

$$S\left(\boldsymbol{m}\right) = Sim\left(\boldsymbol{m}, f_\Phi\left(g_\Theta\left(\boldsymbol{m}\right)\right)\right) = Sim\left(\boldsymbol{m}, \boldsymbol{m}'\right), \tag{1}$$

where $g$ denotes the retrosynthetic planner parameterized by $\Phi$ and $f$ represents the forward model parameterized by $\Theta$.

## 4 EXPERIMENTS

Our experiment consists of two parts. The first part focuses on assessing the reliability of the SA score, search success rate, and round-trip score. The second part evaluates the synthesizability of generated molecules using the round-trip score.

### 4.1 EVALUATING THE RELIABILITY OF SYNTHESIZABILITY METRICS

Currently, retrosynthetic planners are employed to generate synthetic routes for new molecules. When the planner predicts a route, our synthesizability evaluation metrics need to differentiate between feasible and infeasible routes. Therefore, we need a dataset to assess such discriminative capability of these synthesizability evaluation metrics.

**Dataset Construction.** To prepare the dataset, we first clean and deduplicate the USPTO reactions, resulting in approximately 916k reactions. These reactions are then used to construct a reaction network. Molecules with an out-degree of 0 in the network are treated as target molecules, and their corresponding synthetic routes are extracted. This process yields synthetic routes for 107,354 molecules, where the leaf nodes in the routes are starting materials. Note that some molecules can be synthesized through multiple synthetic routes in the dataset.

The dataset is then divided into training, validation, and test sets, based on molecules. These splits consist of 107,154, 100, and 100 data points, respectively. Each data point includes the target molecule and all its associated synthetic routes.

**Settings.** We employ the template-based model Neuralsym as our retrosynthesis model, training it on reactions derived from the 107,154 data points. For predicting synthetic routes for new molecules, we leverage Neuralsym integrated with beam search as our retrosynthetic planner. We use the Transformer Decoder as our forward reaction prediction model, training it on 916,000 reactions. All experiments in this paper are conducted using an Nvidia H100 80G GPU. We find that the primary bottleneck in time complexity during the search is Neuralsym's retrosynthesis prediction, which requires 0.157s per prediction. In contrast, our forward model, which utilizes KV cache and batch decoding, achieves a rapid prediction time of only 0.0055s per reaction.

Table 1: Performance of synthesizability evaluation metrics.

| Metric | Accuracy | Precision | Recall | F1 Score |
|---|---|---|---|---|
| Search Success Rate | - | 71.6% | - | - |
| Round-trip Score | 66.0% | 81.5% | 64.7% | 72.0% |

**Evaluation Protocol.** We use 100 data points from the test set to evaluate the ability of the synthesizability evaluation metric to distinguish between feasible and infeasible routes. For these 100 target molecules, we first employ the retrosynthetic planner to predict synthetic routes with a beam size of 5. During the search process, the depth of each route is restricted to not exceed the maximum depth of the reference route for the target molecule in the test set. While the planner can generate up to five different routes for each molecule, we only consider the route with the highest confidence score. Additionally, our retrosynthetic planner can't generate routes for 5 of the molecules.

To determine the feasibility of a predicted route, we compare it against the reference routes in the test set. If the starting materials of the predicted route match the starting materials of any reference route, the route is deemed feasible. However, it is important to note that the reference routes in the test set do not cover all possible feasible routes. For predicted routes that do not match any reference route, we manually assess their feasibility using the CAS SciFinder (Gabrielson, 2018) tool [2] combined with our domain knowledge.

Through this process, we find that for 56 molecules, the predicted routes are identified as feasible based on the reference routes in the test set. For an additional 12 molecules, feasibility is confirmed through manual evaluation and the use of CAS tools. However, the predicted routes for the remaining 32 molecules are determined to be infeasible. We evaluate the ability of synthesizability evaluation metrics to identify two types of routes: feasible and infeasible.

For **feasible** routes, if the round-trip score is 1, it indicates that our forward reaction model successfully simulates the route to synthesize the target molecule, which is considered a successful identification. For **infeasible** routes, if the round-trip score is not 1, it indicates that the forward reaction model fails to synthesize the target molecule by simulating the route, which is also counted as a successful identification. Based on this, we define the following terms:

- **True Positive (TP)**: Correctly identified feasible routes (round-trip score $= 1$ for actual feasible routes): 44.

- **True Negative (TN)**: Correctly identified infeasible routes (round-trip score $\neq 1$ for actual infeasible routes): 22.

- **False Positive (FP)**: Incorrectly identified feasible routes (round-trip score $= 1$ for actual infeasible routes): 10.

- **False Negative (FN)**: Incorrectly identified infeasible routes (round-trip score $\neq 1$ for actual feasible routes): 24.

Since the search success rate does not evaluate the feasibility of predicted routes, we define these terms as follows:

- **True Positive (TP)**: Feasible routes correctly identified as successful search: 68.

- **False Positive (FP)**: Infeasible routes incorrectly identified as successful search (routes generated but are infeasible): 27.

As the SA scores of these molecules are similar, we conclude that the SA score lacks the ability to differentiate between feasible and infeasible routes. Therefore, we do not include it as a baseline. Additionally, for the search success rate, meaningful TN and FN are absent. Therefore, we use Precision $= \frac{TP}{TP+FP}$ to compare these synthesizability evaluation metrics. Besides, we provide Accuracy $= \frac{TP+TN}{TP+TN+FP+FN}$, Recall $= \frac{TP}{TP+FN}$, and F1 Score $= 2 \times \frac{\text{Precision} \times \text{Recall}}{\text{Precision} + \text{Recall}}$ for round-trip score.

---

[2] https://scifinder-n.cas.org/

Table 2: Performance Comparison of Various Models Using Top-k (Max > 0.9) Route Quality.

| Model | Average Number of Atoms | Ratio of Starting Materials | Top-1 | Top-2 | Top-3 | Top-4 | Top-5 |
|---|---|---|---|---|---|---|---|
| LiGAN | 21.17 | 1.66% | 1.95% | 2.06% | 2.16% | 2.26% | 2.29% |
| DecompDiff | 28.34 | 0.53% | 1.88% | 2.20% | 2.43% | 2.47% | 2.50% |
| TargetDiff | 24.46 | 2.05% | 2.83% | 3.05% | 3.18% | 3.22% | 3.25% |
| DrugGPS | 23.36 | 5.54% | 6.80% | 7.19% | 7.38% | 7.48% | 7.57% |
| AR | 17.98 | 4.67% | 6.75% | 7.32% | 7.58% | 7.83% | 7.96% |
| FLAG | 22.42 | 10.35% | 12.44% | 12.96% | 13.34% | 13.57% | 13.69% |
| Pocket2Mol | 18.53 | 14.75% | 18.08% | 19.03% | 19.44% | 19.56% | 19.78% |

**Results.** Table 1 compares the precision of the search success rate and the round-trip score. The results clearly demonstrate that the round-trip score outperforms the search success rate, emphasizing the strength of the forward reaction model in assessing route feasibility. This is particularly important when minimizing false positives, as inaccurately identifying infeasible routes as feasible can compromise the reliability of the synthesizability evaluation metrics. Moreover, as more reaction data becomes available, the forward reaction model is expected to improve in accuracy, resulting in increasingly reliable round-trip score evaluations.

## 4.2 Benchmarking Generated Molecules with Round-trip Score

In this section, we utilize the round-trip score to assess the synthesizability of molecules generated by SBDD models.

**Settings.** We employ the same forward reaction prediction model and retrosynthesis described in Section 4.1. During the search process, we set the beam size to 5 and limit the depth of each route to a maximum of 15. Due to computational constraints, we are unable to use a beam size of 50 as employed in retrosynthesis evaluation (Dai et al., 2019). Besides, our approach generates about 5 synthetic routes per molecule, in contrast to previous methods in retrosynthetic planning that typically produce only one route. This offers a more comprehensive evaluation compared to the search success rate metric used in earlier studies (Chen et al., 2020) for evaluating search algorithms.

**Baselines.** For our evaluation, we employ a diverse set of state-of-the-art SBDD models, including LiGAN (Ragoza et al., 2022), AR (Luo et al., 2021), Pocket2Mol (Peng et al., 2022), FLAG (Zhang et al., 2022), TargetDiff (Guan et al., 2023a), DrugGPS (Zhang & Liu, 2023), and DecompDiff (Guan et al., 2023b). These models are trained and tested using the CrossDocked dataset (Francoeur et al., 2020), which comprises an extensive collection of 22.5 million protein-molecule structures. Our experimental setup involves randomly selecting 100,000 protein-ligand pairs from this dataset for training purposes. For testing, we draw 100 proteins from the remaining data points. To ensure a comprehensive evaluation, we randomly sample 100 molecules for each protein pocket in the test dataset, resulting in a total of 10,000 molecules. Additionally, we also verify the validity and plausibility of these molecules. After that, we employ our retrosynthetic planner to generate synthetic routes for them.

**Metrics.** We calculate the average number of atoms in the generated molecules and the proportion that are starting materials. Besides, we calculate the percentage of molecules for which at least one of the top-k predicted synthetic routes achieves a round-trip score exceeding 0.9.

**Results.** Based on the results presented in Table 2, we can draw several conclusions. There is a significant variation in performance across different SBDD models. As the average number of atoms in the generated molecules increases, the ratio of starting materials and the top-k performance. The Top-5 Max > 0.9 ranges from 2.29% for LiGAN to 19.78% for Pocket2Mol, indicating a substantial difference in the models' abilities to generate synthetically accessible molecules. Pocket2Mol consistently outperforms other models across all metrics, with 19.78% of its generated molecules having at least one high-quality synthetic route (round-trip score > 0.9) among the top 5 predictions. The improvement in performance from Top-1 to Top-5 suggests that considering multiple top predictions can significantly increase the likelihood of finding feasible synthetic routes.

Notably, the performance ranking of models remains consistent across all top-k evaluations except top-1. The performance increase from Top-4 to Top-5 is less than 1%, indicating that the performance is approaching saturation. Additionally, as shown in Table 4 in Appendix B, the performance gap between models generally widens as $k$ increases, with most gaps showing an upward trend. These observations suggest that our chosen beam size of 5 is sufficient to provide an accurate ranking of each model's performance. This consistency in ranking and the approaching saturation point lend credibility to our evaluation methodology and the reliability of our comparative analysis. More experiment results can be found in Appendix B. The analysis of molecular properties from various generative models, as presented in Table 7, reveals that superior molecular properties do not always correlate with better synthesizability. Even for the best-performing model, a considerable portion of generated molecules still lack high-quality synthetic routes, indicating room for improvement in generating synthetically accessible molecules in SBDD tasks. These findings underscore the importance of evaluating synthetic accessibility in SBDD models and highlight the potential of using top-k predictions to identify feasible synthetic routes for generated molecules.

## 5 RELATED WORK

**Structured-Based Drug Design.** Generative models for Structure-Based Drug Design (SBDD) can be broadly categorized into two main types: non-diffusion and diffusion-based models. Non-diffusion models encompass a range of approaches, including LiGAN (Ragoza et al., 2022), AR (Luo et al., 2021), Pocket2Mol (Peng et al., 2022), GraphBP (Liu et al., 2022b), FLAG (Zhang et al., 2022), and DrugGPS (Zhang & Liu, 2023). On the other hand, diffusion-based models can be considered as a alternative, such as TargetDiff (Guan et al., 2023a), DiffSBDD (Schneuing et al., 2022), and DecompDiff (Guan et al., 2023b).

**Reaction Prediction Model.** Reaction prediction models can be broadly categorized into two approaches: template-based and template-free. Template-based methods (Wei et al., 2016; Segler & Waller, 2017; Qian et al., 2020; Chen & Jung, 2022) begin by extracting reaction templates $\boldsymbol{T} = \{\boldsymbol{T}_1, \ldots, \boldsymbol{T}_{N(T)}\}$ from a reaction database. These methods then predict the most suitable template class based on the given reactants and apply the predefined, encoded template to generate the product. Template-free approaches, on the other hand, are more diverse. Some, inspired by chemical reaction mechanisms, adopt a two-stage learning process (Jin et al., 2017). They first identify the chemical reaction centers of the reactants using atom mapping numbers, and then form new bonds or break existing ones between atoms at these centers. However, most contemporary strategies employ an end-to-end, template-free learning paradigm for reaction prediction. Several methods (Yang et al., 2019; Schwaller et al., 2019a; Tetko et al., 2020; Irwin et al., 2022; Lu & Zhang, 2022; Zhao et al., 2022; Tu & Coley, 2022) frame reaction prediction as a sequence-to-sequence or graph-to-sequence problem. Other approaches (Bradshaw et al., 2019; Do et al., 2019; Sacha et al., 2021; Bi et al., 2021; Meng et al., 2023) predict the product by directly performing graph transformations on the reactants' graph representations.

**Retrosynthesis Model.** Existing retrosynthesis models (Segler & Waller, 2017; Coley et al., 2017; Liu et al., 2017; Zheng et al., 2019; Chen et al., 2019; Dai et al., 2019; Karpov et al., 2019; Chen et al., 2020; Ishiguro et al., 2020; Guo et al., 2020; Tetko et al., 2020; Shi et al., 2020; Yan et al., 2020; Seo et al., 2021; Chen & Jung, 2021; Lee et al., 2021; Seidl et al., 2021; Somnath et al., 2021; Sun et al., 2021; Yan et al., 2022; Fang et al., 2022; Gao et al., 2022b; Wan et al., 2022; He et al., 2022; Liu et al., 2022a; Tu & Coley, 2022; Lin et al., 2022; Zhong et al., 2022; Baker et al., 2023; Zhong et al., 2023; Yu et al., 2023; Li et al., 2023b; Xie et al., 2023; Sacha et al., 2023; Zhu et al., 2023b; Jiang et al., 2023; Qian et al., 2023; Xiong et al., 2023; Wang et al., 2023; Gao et al., 2023; Zhu et al., 2023a; Lan et al., 2023; Chen et al., 2023; Yao et al., 2023; Lin et al., 2023; Liu et al., 2024b; Zhang et al., 2024b;a; Lan et al., 2024) can be broadly categorized into three main types: template-free, semi-template-based, and template-based methods. These categories can be further refined based on their utilization of atom mapping information. Template-free methods typically approach retrosynthesis as either a translation problem (Karpov et al., 2019) or a graph edit problem (Sacha et al., 2021). Some of these methods optionally use atom mapping to align input and output molecules (Seo et al., 2021; Zhong et al., 2022; Yao et al., 2023). Template-based methods (Segler & Waller, 2017; Dai et al., 2019) leverage atom mapping information to create a pool of reaction templates. They generally frame retrosynthesis as a template classification or retrieval problem.

Semi-template-based methods (Shi et al., 2020; Somnath et al., 2021) also utilize atom mapping information, but they employ it to obtain other prior information, such as identifying reaction centers. Many of these methods adopt a two-stage learning paradigm for retrosynthesis: First, they identify the reaction center in the product molecule and break it into synthons. Then, they transform these synthons into reactants.

**Search Algorithm.**   A variety of search algorithms have been developed to navigate the synthetic planning. These include beam search, neural A* search (Chen et al., 2020; Han et al., 2022; Xie et al., 2022), Monte Carlo Tree Search (MCTS) (Segler et al., 2018; Hong et al., 2021), and reinforcement learning-based (RL-based) search (Yu et al., 2022). Some other works to this field include (Kishimoto et al., 2019; Heifets & Jurisica, 2012; Kim et al., 2021; Hassen et al., 2022; Li et al., 2023a; Zhang et al., 2023; Liu et al., 2023a; Lee et al., 2023; Yuan et al., 2024; Tripp et al., 2024). These algorithms are designed to explore the vast reaction space, prioritizing the most promising synthetic routes.

## 6  CONCLUSION

In this work, we propose a novel round-trip score to assess the synthesizability of molecules generated by existing SBDD models. This score evaluates the likelihood that a retrosynthetic planner, trained on current reaction data, can predict feasible synthetic routes for these molecules. To enhance the robustness of our evaluation method, we advocate for the release of additional reaction data. This expanded dataset would significantly improve the accuracy and reliability of our assessments.

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

# A  REPRODUCIBILITY

We use Pytorch (Paszke et al., 2019) to implement our retrosynthesis and reaction prediction models. The softwares that we use for experiments are Python 3.6.8, CUDA 10.2.89, CUDNN 7.6.5, einops 0.4.1, pytorch 1.9.0, pytorch-scatter 2.0.9, pytorch-sparse 0.6.12, numpy 1.19.2, torchvision 0.10.0, and torchdrug 0.1.3.

Table 3: The hyper-parameters for the reaction model.

| | |
|---|---|
| max length | 402 |
| embedding size | 64 |
| decoder layers | 6 |
| attention heads | 8 |
| FFN hidden | 2048 |
| dropout | 0.1 |
| epochs | 2000 |
| batch size | 128 |
| warmup | 16000 |
| lr factor | 20 |
| scheduling | $lr = \frac{\text{lr factor} \times \min\left(1.0, \frac{0.1 \, \text{num\_step}}{\text{warmup}}\right)}{\max(0.1 \, \text{num\_step}, \text{warmup})}$ |

Table 3 reports the hyper-parameter setting of our reaction model. For Neuralsym, we follow the setting in `https://github.com/linminhtoo/neuralsym`.

# B  ADDITIONAL EXPERIMENTAL RESULTS

## B.1  SYNTHESIZABILITY EVALUATION

Tables 4 and 6 present performance gaps between consecutive models for top-k metrics with thresholds of 0.9 and 0.8, respectively. Table 5 compares model performance using top-k route quality with a threshold of 0.8.

Table 4: Performance Gaps (%) Between Consecutive Models for Top-k (Max > 0.9) Metrics

| Gap Between Models | Top-1 | Top-2 | Top-3 | Top-4 | Top-5 |
|---|---|---|---|---|---|
| LiGAN to DecompDiff | -0.07 | 0.14 | 0.27 | 0.21 | 0.21 |
| DecompDiff to TargetDiff | 0.95 | 0.85 | 0.75 | 0.75 | 0.75 |
| TargetDiff to DrugGPS | 3.97 | 4.14 | 4.20 | 4.26 | 4.32 |
| DrugGPS to AR | -0.05 | 0.13 | 0.20 | 0.35 | 0.39 |
| AR to FLAG | 5.69 | 5.64 | 5.76 | 5.74 | 5.73 |
| FLAG to Pocket2Mol | 5.64 | 6.07 | 6.10 | 5.99 | 6.09 |

Table 5: Performance (%) Comparison of Various Models Using Top-k Route Quality (Max > 0.8).

| Model | Top-1 | Top-2 | Top-3 | Top-4 | Top-5 |
|---|---|---|---|---|---|
| LiGAN | 2.05 | 2.16 | 2.27 | 2.37 | 2.42 |
| DecompDiff | 2.36 | 2.77 | 3.04 | 3.18 | 3.27 |
| TargetDiff | 2.86 | 3.09 | 3.22 | 3.27 | 3.31 |
| DrugGPS | 7.03 | 7.46 | 7.68 | 7.81 | 7.92 |
| AR | 6.96 | 7.69 | 7.99 | 8.28 | 8.46 |
| FLAG | 12.74 | 13.40 | 13.85 | 14.13 | 14.33 |
| Pocket2Mol | 18.66 | 19.77 | 20.27 | 20.44 | 20.65 |

The properties of generated molecules presented in Table 7 are derived from two primary sources. For LiGAN, AR, Pocket2Mol, FLAG, and DrugGPS, all reported metrics (Vina Score, High Affinity, QED, SA, LogP, Lip., Sim. Train, and Div.) are extracted from the DrugGPS paper. For TargetDiff and DecompDiff, the Vina Score, High Affinity, QED, SA, and Div. metrics are sourced from the DecompDiff paper.

Table 6: Performance Gaps (%) Between Consecutive Models for Top-k (Max > 0.8) Metrics

| Gap Between Models | Top-1 | Top-2 | Top-3 | Top-4 | Top-5 |
|---|---|---|---|---|---|
| LiGAN to DecompDiff | 0.31 | 0.61 | 0.77 | 0.81 | 0.85 |
| DecompDiff to TargetDiff | 0.50 | 0.32 | 0.18 | 0.09 | 0.04 |
| TargetDiff to DrugGPS | 4.17 | 4.37 | 4.46 | 4.54 | 4.61 |
| DrugGPS to AR | -0.07 | 0.23 | 0.31 | 0.47 | 0.54 |
| AR to FLAG | 5.78 | 5.71 | 5.86 | 5.85 | 5.87 |
| FLAG to Pocket2Mol | 5.92 | 6.37 | 6.42 | 6.31 | 6.32 |

An analysis of Table 7 reveals a crucial insight: superior molecular properties do not necessarily translate to higher round-trip scores or search success rates. This observation underscores a critical aspect of molecular generation in drug discovery - the importance of balancing molecular quality with synthesizability. While generating high-quality molecules is essential, ensuring that these molecules are practically synthesizable is equally crucial for advancing potential drug candidates. This finding highlights the need for a holistic approach in generative models for drug discovery, one that considers both the desirable properties of molecules and their feasibility for synthesis.

Table 7: Comparing the generated molecules' properties by different generative models. We report the means and standard deviations. The properties of the test dataset for the best results are bolded.

| Model | Vina Score (kcal/mol, ↓) | High Affinity(↑) | QED (↑) | SA (↑) | LogP | Lip. (↑) | Sim. Train (↓) | Div. (↑) |
|---|---|---|---|---|---|---|---|---|
| LiGAN | $-6.03_{\pm 1.89}$ | $0.19_{\pm 0.26}$ | $0.37_{\pm 0.27}$ | $0.62_{\pm 0.20}$ | $-0.02_{\pm 2.48}$ | $4.00_{\pm 0.92}$ | $0.41_{\pm 0.22}$ | $0.67_{\pm 0.15}$ |
| AR | $-6.11_{\pm 1.66}$ | $0.24_{\pm 0.23}$ | $0.48_{\pm 0.18}$ | $0.66_{\pm 0.19}$ | $0.21_{\pm 1.76}$ | $4.69_{\pm 0.45}$ | $0.39_{\pm 0.21}$ | $0.65_{\pm 0.13}$ |
| Pocket2Mol | $-6.87_{\pm 2.19}$ | $0.41_{\pm 0.23}$ | $0.52_{\pm 0.24}$ | $0.73_{\pm 0.21}$ | $0.83_{\pm 2.17}$ | $4.89_{\pm 0.22}$ | $0.36_{\pm 0.19}$ | $0.70_{\pm 0.17}$ |
| TargetDiff | -5.47 | | 0.58 | 0.48 | 0.58 | - | - | 0.72 |
| FLAG | $-6.96_{\pm 1.92}$ | $0.45_{\pm 0.22}$ | $0.55_{\pm 0.20}$ | $0.74_{\pm 0.19}$ | $0.75_{\pm 2.09}$ | $4.90_{\pm 0.14}$ | $0.39_{\pm 0.18}$ | $\mathbf{0.70_{\pm 0.18}}$ |
| DecompDiff | -5.67 | **0.64** | 0.45 | 0.61 | - | - | - | 0.68 |
| DrugGPS | $\mathbf{-7.28_{\pm 2.14}}$ | $0.57_{\pm 0.23}$ | $\mathbf{0.61_{\pm 0.22}}$ | $\mathbf{0.74_{\pm 0.18}}$ | $0.91_{\pm 2.15}$ | $\mathbf{4.92_{\pm 0.12}}$ | $\mathbf{0.36_{\pm 0.21}}$ | $0.68_{\pm 0.15}$ |

