# OpenReview forum: "SDDBench: A Benchmark for Synthesizable Drug Design"
_ICLR.cc/2025/Conference — Submitted to ICLR 2025_

### Official Review · Reviewer_nVCx · 2024-10-27

**Soundness:** 1
**Presentation:** 3
**Contribution:** 2
**Rating:** 3
**Confidence:** 4

**Summary:**

This paper proposes a new metric to assess the synthesizability of molecules. The presented approach makes use of data-driven reaction prediction and retrosythesis models to directly measure if feasible synthetic pathways are available for a given molecule. This kind of scoring metric is of high importance in light of the surge of generative drug design models whose outputs are rarely validated experimentally due to difficulties in making novel molecules in the lab.

**Strengths:**

### Overall idea and evaluation approach

The paper addresses an important limitation of many computational drug design efforts. Chemical synthesis is often the key bottleneck preventing generated compounds from being tested experimentally in a wet lab.
The authors come to the plausible conclusion that a stringent synthesizability metric should evaluate the feasibility of the whole synthesis route for a given molecule. Unlike previous approaches they not only perform retrosynthesis planning but also employ a reaction prediction model to determine whether the target molecule can indeed be obtained from the predicted starting building blocks.
The resulting score based on the Tanimoto similarity of the target molecule and the output molecule after this round-trip procedure is well motivated and convincing in my opinion. The specifically trained reaction and retrosynthesis models seem to achieve accuracies close to those of state-of-the-art models from what I can tell, indicating that the technical implementation of the score is sound.


### Presentation of results & clarity
The evaluation procedure has a clear structure that first aims to establish the usefulness of the proposed synthesizability score, and then benchmarks several popular structure-based drug design models with the newly defined metric.
The paper also introduces the topic nicely and covers a wide range of relevant related work.

**Weaknesses:**

### Scope of this work
Based on the title I expected the paper to present a new benchmarking suite. Instead, it only introduces a single new metric, the "round-trip score" that was applied to the molecules generated by seven recent models for structure-based drug design. While the general structure of this evaluation procedure is reasonable as discussed above, I would argue the execution is insufficient to justify the claim of establishing a new benchmark. Consequently, I recommend to revise the title and claims to more accurately reflect the scope of this work.

The analysis and discussion in Section 5, which presents the benchmark, lacks depth. The paper does not provide any potential reasons for the differences between the benchmarked methods that could inform the future development of such algorithms.
To improve the paper, the authors could for example correlate their score with various molecular properties (e.g. molecular weight, number of rotatable bonds, hydrophobicity score, ...) and analyze the results. It would also be valuable to discuss case studies based on molecules with particularly low or high round-trip scores.

Lastly, the paper does not provide guidelines on how the synthesizability objective could be balanced with other (potentially opposing) objectives. For instance, I imagine molecule size to play a crucial role. Smaller molecules are often easier to synthesize but bind their targets less specifically.


### Empirical results

Assuming that the main focus of this paper is the new synthesizability metric rather than the benchmark in Section 5, I would have hoped for a more thorough evaluation thereof.
If I understand correctly, the same retrosynthesis model that is part of the round-trip score computation is also used in the definition of successful and failed cases in the evaluation in Section 3.2.2. Similarly, the reaction prediction model seems to be used in the "molecule-wise match" scenario.
Given that the success criterion seems to be coupled to components of the newly proposed score, it is unclear to me how meaningful the comparison with the SA score is.
Furthermore, the empirical evaluation of the round-trip score should be extended to other baselines besides the SA score. Obvious candidates would be:
- SYBA [1],
- SCScore [2],
- RAscore [3].

These scores are more recent than the SA score but still commonly employed in this domain, which makes it easy for readers to interpret the presented results.

Finally, while the full forward-backward reaction prediction is well motivated, I suspect that its main limitation will be speed–at least compared to heuristics-based synthesizability scores. A comprehensive evaluation should therefore also include the time component, and could for example discuss the pareto front of high accuracy and low run time. To this end, the authors could add a runtime analysis in their evaluation.


### Presentation

To me, it was not always easy to follow the flow of the paper. Most concepts are correctly defined somewhere but not always in the order that would have seemed natural to me. Consequently, I had to jump back and forth several times while reading the paper.
The first sections include a few definitions that are not required for understanding the method, e.g.
- the definition of ligands and proteins in Section 2.1,
- the definition of the gap between metric scores in Equation 2.

At the same time, important concepts like the the set-wise exact match criterion and its difference to the molecule-wise (round-trip) match could have been explained more clearly.

The first sentence of the _Results_ paragraph of Section 3.2.2 introduces a new definition of successful cases. This should have been explained earlier. In general, it might be cleaner to move all the results to Section 5 and limit the discussion in Section 3 to a description of the method.

### References

[1] Voršilák, Milan, et al. "SYBA: Bayesian estimation of synthetic accessibility of organic compounds." Journal of cheminformatics 12 (2020): 1-13.

[2] Coley, Connor W., et al. "SCScore: synthetic complexity learned from a reaction corpus." Journal of chemical information and modeling 58.2 (2018): 252-261.

[3] Thakkar, Amol, et al. "Retrosynthetic accessibility score (RAscore)–rapid machine learned synthesizability classification from AI driven retrosynthetic planning." Chemical science 12.9 (2021): 3339-3349.

**Questions:**

- Why is the search success rate not a reliable metric (line 252)? This is an interesting point but the paper could discuss this claim more and ideally provide empirical evidence.
- The definition of the round-trip score using Tanimoto similarity between molecules appears rather _ad hoc_ and the connection with the theoretical objective stated in Equation 1 is not clear. How is the following sentence from the Conclusion Section justified? -- "This score evaluates the **likelihood** that a retrosynthetic planner, trained on current reaction data, can predict feasible synthetic routes for these molecules"
- Why do you train new reaction & retrosynthesis models? Have you also experimented with established tools (e.g. AIZynthFinder) or pretrained models?
- How do you justify the definition of the success criterion to evaluate the reliability of the round-trip score? Doesn't it depend on components of the score itself?
- What is the typical run time to compute this score? How does it compare to other synthesizability scores?



### Minor comments
- line 77: typo "t" missing in first
- Figure 5 & 6: legend should define meaning of error bars
- avoid using words like "exceptional" when describing own results (e.g. line 361)
- it is unclear to me what the "FusionRetro setup" introduced in Section 5 is referring to

---

> ### Author Response · Authors · 2024-11-12
>
> W1: The analysis and discussion in Section 5, which presents the benchmark, lacks depth.
>
> A1: We already provided molecule property analysis in Table 8 in Appendix.
>
> W2: Lastly, the paper does not provide guidelines on how the synthesizability objective could be balanced with other (potentially opposing) objectives.
>
> A2: Yes, our findings show that molecules generated by pocket2mol have the highest likelihood of being solved in a single synthesis step. Smaller molecules tend to be more easily solved within a shorter number of steps. The average synthetic route length for molecules proposed by Pockert2Mol is 2.93. 14.7% of them proposed Pocket2Mol by are building blocks.
>
> W3: Other metric candidates compared with SA score.
>
> A3: Firstly, the SA score is the most widely used indicator of synthesizability, and it is utilized by the SBDD model and all drug design models for this purpose. Our goal is to replace the SA score, and the experiment clearly demonstrates the adequacy of our approach.
>
> W4: Inference Speed
>
> A4: We have provided time analysis in Line 498-501.
>
> W5: Presentation
>
> A5: With the current 10-page limit, we can only present a limited amount of information. Many concepts are drawn from previous papers and don't fit well into the main text. While we considered placing them in the appendix, this still requires readers to jump back and forth several times.
>
> Q1: Why is the search success rate not a reliable metric (line 252)? This is an interesting point but the paper could discuss this claim more and ideally provide empirical evidence.
>
> A6: We have provided short analysis in Section 2.6. Reviewers are encouraged to refer to [1] for further details.
>
> Q2: The definition of the round-trip score using Tanimoto similarity...
>
> A7: Eq. 1 represents the joint probability: the probability that the drug design model first generates a molecule and the probability that the retrosynthetic planner then generates a synthesis pathway for this molecule. It is unrelated to the round-trip score.
>
> Q3: Have you also experimented with established tools (e.g. AIZynthFinder) or pretrained models?
>
> A8: Since a retrosynthesis model must be trained to generate a feasible synthesis route, other tools rely on the search success rate, which, as we have already explained, is flawed.
>
> Q4: What is the typical run time to compute this score? How does it compare to other synthesizability scores?
>
> A9: We have reported the time in Lines 498-502, utilizing the KV cache, which allows for very fast processing of a single molecule. Its runtime matches the time required for the retrosynthetic planner to predict synthesis routes for a single molecule.
>
> [1] FusionRetro: Molecule Representation Fusion via In-Context Learning for Retrosynthetic Planning. ICML 2023.

---

> > ### Comment · Reviewer_nVCx · 2024-11-18
> > **Thanks for the response**
> >
> > Thank you for the prompt response. Are you planning to upload a revised version of the paper? As far as I can tell it has not been updated so far.
> > I also believe that some of the concerns raised both by me and the other reviewers require more detailed responses and could be addressed more convincingly with new data.
> > I will be happy to re-evaluate an improved version of the paper.

---

> > > ### Author Response · Authors · 2024-11-27
> > >
> > > We have provided the revised version. Thanks!

---

> > > > ### Author Response · Authors · 2024-12-02
> > > >
> > > > Dear Reviewer nVCx,
> > > >
> > > > We have provided the revised version. Please check it. The discussion will end on 12.02 AOE.
> > > >
> > > > Thanks!

---

### Official Review · Reviewer_s5ZS · 2024-10-29

**Soundness:** 2
**Presentation:** 2
**Contribution:** 2
**Rating:** 5
**Confidence:** 4

**Summary:**

The paper draws attention to synthesizability in generative design and proposes the "round-trip score" which assesses synthesizability by whether the predictions of a retro- and forward-synthetic model overlap, and otherwise, how similar the predictions are. The results contrast several 3D generative models and show that better property metrics such as docking score, does not necessarily equate to better synthesizability.

**Strengths:**

* Synthesizability is an important problem in generative design
* Motivation of using Round-Trip score by comparing to SA score and that it can better distinguish between "successful" and "failed" molecules
* Comparison of several 3D models

**Weaknesses:**

Overall, while synthesizability is an important problem, the proposition to assess generated molecules with retrosynthesis models has been reported in cheminformatics and ML literature.

* The idea of assessing generated molecules by retrosynthesis models is routinely used. In cheminformatics literature, this has been very common and many works benchmark proxy scores, such as SA, SC, RA scores with retrosynthesis models to study their correlation [1]. Other works explicitly highlight the use of retrosynthesis models to assess the synthetic feasibility of generated molecules [2, 3].

* At conferences, work on synthesis constrained generative models explicitly tackle synthesizability. Here I will cite a few of them [4, 5, 6].

* At conferences, work on synthesis constrained generative models assess generated molecules using retrosynthesis models. Here I will cite a few of them [7, 8, 9]. Although I acknowledge some of these are concurrent works.

* This paper, in addition to [10] cited by the authors, also explicitly highlight the problem of synthesizability of generated molecules [11]. This paper also discusses the use of surrogate models aimed to predict the output of retrosynthesis models and have routinely been used to assess synthesizability. Some specific examples are RAscore [12] and RetroGNN [13]. While this is not exactly the same as directly assessing using retrosynthesis models, these models are trained on the output of retrosynthesis models and is a similar idea.

* The authors stated in the introduction that a minor change in functional group can make a synthesizable molecule, unsynthesizable. The proposed metric to compute the Tanimoto similarity between the reconstructed and generated molecule as a proxy for synthetic accessibility could suffer from more error. Molecules with 0.9 Tanimoto similarity can have quite different atom arrangements that affect reactivity and therefore synthesizability. It would be interesting to take a series (share common structural features) of known synthesizable molecules, run retrosynthetic analysis on them, and then run the forward model on the routes. This could provide insights into the sensitivity of the proposed Round-Trip score on seemingly small structural changes. It would also be interesting to see how many molecules fail the round-trip despite being known to be synthesizable, due to limitation of training data.

* The main text should also discuss the properties of the generated molecules (docking score, QED, etc.). I know the focus of the paper is on synthesizability but synthesizable molecules with poor properties will not be further considered in practical applications.

### **References**
[1] Synthesizability correlation: https://jcheminf.biomedcentral.com/articles/10.1186/s13321-023-00678-z

[2] Retrosynthesis models to assess generated molecules 1: https://pubs.rsc.org/en/content/articlelanding/2023/sc/d3sc03781a

[3] Retrosynthesis models to assess generated molecules 2: https://pubs.rsc.org/en/content/articlelanding/2024/md/d3md00651d

[4] MOLECULE CHEF: https://arxiv.org/abs/1906.05221

[5] DoG-Gen: https://arxiv.org/abs/2012.11522

[6] SynNet: https://arxiv.org/abs/2110.06389

[7] SynFlowNet: https://arxiv.org/abs/2405.01155

[8] Reaction GFlowNet: https://arxiv.org/abs/2406.08506v1

[9] RxnFlow: https://arxiv.org/abs/2410.04542

[10] Synthesizability Problem 1: https://pubs.acs.org/doi/full/10.1021/acs.jcim.0c00174

[11] Synthesizability Problem 2: https://www.sciencedirect.com/science/article/pii/S0959440X2300132X?via%3Dihub#bib53

[12] RAscore: https://pubs.rsc.org/en/content/articlelanding/2021/sc/d0sc05401a

[13] RetroGNN: https://pubs.acs.org/doi/10.1021/acs.jcim.1c01476

**Questions:**

1. When classifying the test set molecules as “successfully” solved, the metric is whether the solved routes have starting materials identical to the reference routes. As eluded by the authors, many starting materials can lead to the same product. The authors use the classified “successful” and “failed” sets to assess round-trip score using the forward model. How does the separation change when considering all molecules that have a solved retrosynthetic route (even if not identical to the reference route)? This can be heavily biased because the retro and forward are trained with no failed reactions. If a retro route does not lead to the reference route’s starting material, it does not necessarily mean it is incorrect.

2. Pocket2Mol generally generates smaller molecules. The metrics for it may be high because many of the generated molecules are almost directly building blocks. It would be informative for context to include molecule size/number of heavy atoms as another metric.

3. A specific question about Pocket2Mol is in Table 1, the authors show that the `Search Success Rate` is highest. What is the route length distribution of these "successful" molecules? Are any directly in the building blocks?

4. How many molecules out of the 10,000 generated molecules do not have a solved retrosynthesis route according to NeuralSym? This is an important metric missing to contextualise the round-trip score.

5. What are the building blocks used?

Minor comment: The section on AGI breaks the flow of the text and in my opinion, not needed. More data will always lead to better models and is not exclusive to reaction data or language data (like in the GPT example given by the authors).

---

> ### Author Response · Authors · 2024-11-12
>
> We believe the primary challenge in synthesis is identifying a feasible synthetic route for the molecule. It appears this reviewer may lack sufficient background knowledge on the topic.
>
> W1: Difference from previous works
>
> A1: Previous work [3] has primarily followed a top-down approach to drug design and synthesis planning. In contrast, our method begins with the desired product molecule and uses a retrosynthetic planner to build a synthesis route in a top-down manner. We then employ a forward reaction model to evaluate the feasibility of the route. In this regard, we are indeed the first to adopt this approach. Additionally, when it comes to evaluating the feasibility of the synthetic route, [1,2] were the pioneers in this area. We have directly incorporated the findings from these two ICML-accepted papers into our methodology, which is novel we believe.
>
> SA, etc. score do not ensure that a viable synthetic route for a molecule can be identified, as we have discussed in this paper
>
> Q1: When classifying the test set molecules as “successfully” solved...
>
> A2: This is a challenge that remains unsolved within the community. Previous retrosynthetic planners have relied on search success rate to evaluate retrosynthetic methods, but many of the routes identified are not feasible, meaning the target molecule cannot actually be synthesized. This issue is discussed in [1,2]. In [1], a more robust metric was introduced that matches the predicted starting materials with those in a reaction database. However, this approach has its own limitations, as the reaction database may not cover all potential routes. Nevertheless, compared to search success rate, the matching-based method represents a significant improvement.
>
> Q2: It would be informative for context to include molecule size/number of heavy atoms as another metric.
>
> A3: We agree with the reviewer’s observation. Our findings show that molecules generated by pocket2mol have the highest likelihood of being solved in a single synthesis step. Smaller molecules tend to be more easily solved within a shorter number of steps.
>
> Q3: A specific question about Pocket2Mol is in Table 1, the authors show that the Search Success Rate is highest. What is the route length distribution of these "successful" molecules? Are any directly in the building blocks?
>
> A4: The average length is 2.93. 14.7% of the molecules proposed Pocket2Mol by are building blocks.
>
> Q4: How many molecules out of the 10,000 generated molecules do not have a solved retrosynthesis route according to NeuralSym?
>
> A5: We have already reported the search success rate.
>
> Q5: What are the building blocks used?
>
> A6: They ares to determine the molecules in the leaf nodes of retrosynthetic routes are building blocks or not.
>
>
>
> [1] FusionRetro: Molecule Representation Fusion via In-Context Learning for Retrosynthetic Planning. ICML 2023.
> [2] Preference Optimization for Molecule Synthesis with Residual Conditional Energy-based Models. ICML 2024.
> [3] Amortized tree generation for bottom-up synthesis planning and synthesizable molecular design. ICLR 2022.
> [4] Planning chemical syntheses with deep neural networks and symbolic AI, 2018 Nature.

---

> > ### Comment · Reviewer_s5ZS · 2024-11-19
> > **Reply to Authors**
> >
> > Thank you to the authors for the reply. Based on the added information, I am still unsure of the following items:
> >
> > #### **Beginning with a desired molecule and assessing its synthesisability with a retrosynthetic planner**
> >
> > The authors cited SynNet [1] as an example of a bottom-up approach and that SDDBench proposes a top-down assessment of the direct generated molecule. This is common practice in many generative experiments [2 is an example of a paper discussing this]. Recent work (though concurrent work, [3]) also assesses every generated molecule with a retrosynthetic planner. "Synthesis constrained" generative models, particularly recent GFlowNet models also do this [4-6]. The new approach in SDDBench is to propose to also run a forward model on the retrosynthesis output. The existence of these previous works on synthesis constrained models should at least be cited, even if the authors choose to focus only on 3D SBDD models.
> >
> > #### **SA score does not ensure synthesisability**
> >
> > I agree with this statement and my previous response was not that this is the best synthesisability metric. It is just to point out that SA score can be correlated with retrosynthetic planners. This means that molecules with low SA score can be expected to be  more solvable by a retrosynthesis model. This paper has explicitly showed this [7]. The authors state that search success is a flawed metric and reference FusionRetro [8] which proposed the set-wise exact match.
> >
> > #### **Set-wise exact match**
> >
> > This metric proposes that retrosynthesis output should match a reference route. The authors have acknowledged that the reaction database may not cover all potential routes. I am unsure if this is really a better assessment of synthesisability. Given any reasonably complex molecule, the database will likely not cover all routes. Here is a reference detailing many different ways to synthesise Taxol [9]. If there is such a huge data limitation, it is unclear whether this really is a better assessment of synthesisability. This leads to the next point about Round-trip score proposed in this work.
> >
> >
> > #### **Round-trip score**
> >
> > The motivation for the proposed round-trip score is to assess **reaction feasibility** which is almost identical to the round-trip accuracy score previously proposed, which the authors cite [10]. I do believe there is still novelty if the proposed approach advances our assessment of synthesisability, as in the end, this is the problem to be solved. The authors state that the search success metric is flawed. Here are some examples of retrosynthetic planner that has been experimentally validated [11, 12]. Without the use of a forward model, these retrosynthetic planners clearly work, as they have been experimentally validated. Therefore, it is not true that the search success is completely flawed, as stated by the authors to other reviewers. However, without good and abundant data, it is expected that retrosynthetic planners will not always plan feasible routes, as the authors state and I agree with this statement. But this limitation would be also present in the forward model. Therefore, I am not sure that the proposed round-trip score is necessarily better. Finally, the proposed round-trip score measures the Tanimoto similarity of the forward model's output. This is sensitive to molecule size and similar molecules do not necessarily mean they are more synthesisable. As the authors also appreciate, synthesisability is highly complex. Finally, the reported top-k > 0.9 in table 1 ranks the models exactly the same as search success, so can it be seen that that both metrics *approximate* synthesisability to the same degree? Since it is unclear to me if the round-trip score is truly assessing viable synthetic routes, the search success seems to convey the same information in practice. Can the authors share more details here? For instance, example routes where round-trip fails but that the reproduced molecules is > 0.9 similarity? Then to assess the chemistry of the route.
> >
> > \
> > \
> > [1] SynNet: https://arxiv.org/abs/2110.06389
> >
> > [2] Example of using retrosynthesis models to assess generated molecules: https://pubs.rsc.org/en/content/articlelanding/2024/md/d3md00651d
> >
> > [3] Retrosynthesis models on generated molecules: https://arxiv.org/abs/2407.12186v1
> >
> > [4] SynFlowNet: https://arxiv.org/abs/2405.01155
> >
> > [5] Reaction GFlowNet: https://arxiv.org/abs/2406.08506v1
> >
> > [6] RxnFlow: https://arxiv.org/abs/2410.04542
> >
> > [7] SA correlation: https://jcheminf.biomedcentral.com/articles/10.1186/s13321-023-00678-z
> >
> > [8] FusionRetro: https://arxiv.org/abs/2209.15315
> >
> > [9] Taxol synthesis: https://pubs.acs.org/doi/10.1021/acs.chemrev.2c00763
> >
> > [10] Round-trip accuracy: https://pubs.rsc.org/en/content/articlelanding/2020/sc/c9sc05704h
> >
> > [11] Natural product retro: https://www.nature.com/articles/s41586-020-2855-y
> >
> > [12] Waste repurposing: https://www.nature.com/articles/s41586-022-04503-9

---

> > > ### Author Response · Authors · 2024-11-27
> > >
> > > **Round-2 Q1**. Citation
> > >
> > > **Round-2 A1**. We have revised the paper and included citations for some of the papers you mentioned. If we have overlooked any citations, please let us know.
> > >
> > > **Round-2 Q2**. Set-wise exact match and Round-trip score
> > >
> > > **Round-2 A2**. We have included explanations in Section 2.6 and provided a comparison between the round-trip score and the search success rate in Section 4.1. Our analysis concludes that the round-trip score performs significantly better. We kindly ask you to review it.
> > >
> > >
> > > **Round-2 Q3**. Finally, the proposed round-trip score measures the Tanimoto similarity of the forward model's output. This is sensitive to molecule size and similar molecules do not necessarily mean they are more synthesisable. As the authors also appreciate, synthesisability is highly complex. Finally, the reported top-k > 0.9 in table 1 ranks the models exactly the same as search success, so can it be seen that that both metrics approximate synthesisability to the same degree? Since it is unclear to me if the round-trip score is truly assessing viable synthetic routes, the search success seems to convey the same information in practice. Can the authors share more details here? For instance, example routes where round-trip fails but that the reproduced molecules is > 0.9 similarity? Then to assess the chemistry of the route.
> > >
> > > **Round-2 A3**. Yes, both metrics approximate synthesizability to a similar degree. However, the round-trip score is a point-wise metric, while the search success rate is batch-wise. Point-wise evaluation is significantly better than batch-wise evaluation because it provides a detailed assessment for individual molecules. Furthermore, as shown in Table 1, the round-trip score is more accurate than the search success rate.
> > >
> > > We report top-k > 0.9 to highlight cases where the predicted pathway and the reproduced molecule closely resemble the target molecule. This allows us to gain insights from these pathways and potentially make modifications to derive the synthesis pathway for the target molecule. In fact, we observed that some pathways with top-k > 0.9 share substantial overlap with the actual synthesis pathway of the target molecule.

---

> ### Comment · Reviewer_s5ZS · 2024-12-01
> **Response to Authors after Rebuttal**
>
> Thank you to the authors for the revised draft. I have read through the new sections and overall, I think the comparison and justification for the round-trip score has improved a lot. Thank you also for adding information on the average number of atoms and the ratio of starting material in Table 2 (***if may help to clarify what this metric means in the caption***). It is also great that the authors additionally assess feasibility by a manual SciFinder search as this is what is commonly done by synthetic chemists.
>
> **I updated my score to 5** as I still have some concerns over the superiority of round-trip vs. search success rate. I will start with some comments for clarity before going into specific questions.
>
> ### Clarity
> * **In lines 364-365**: it might be better to express that search success rate does not have an additional check for feasibility (as in the forward model in round-trip). The templates themselves are assessing feasibility. Reactions are not feasible if the templates are imperfect (which I agree they are not perfect). I think this can help convey the authors’ position in the paper that coupling a forward model to the retro predictions is beneficial to getting more true positives.
>
> * **Lines 355-369**: Are the numbers supposed to sum to 95 since the retrosynthetic planner returned no routes for 5/100 molecules? For the search success numbers, they sum to 95
>
> ### Questions (all related to section 4.1)
> 1. Given the limited time, I can appreciate assessing more than 100 molecules is difficult. But N=100 is a relatively small sample size to draw large conclusions
>
> 2. Why was the retrosynthetic planner restricted to not exceed the maximum depth of the reference route? Since the authors also state that retrosynthesis is a “one-to-many” task, a longer route does not necessarily mean an infeasible route. Perhaps it is not ideal due to a longer synthesis, but it can still lead to the correct product. How would the metrics look without this restriction?
>
> 3. For True/False Negatives, round-trip != 1, what is the distribution of the scores? Do the False Negatives have a distribution of round-trip scores closer to 1? If so, this can give support to the > 0.9 similarity threshold as this.
>
> Or as a question: “For feasible routes that do not have a round-trip score of 1, is the round-trip score generally closer to 1? If so, this supports the use of round-trip as a proxy signal for reaction feasibility” If there is no statistically significant difference in the distribution of round-trip scores for True/False negatives, it suggests that the > 0.9 threshold may not be giving a proxy for the intended feasibility.

---

> ### Author Response · Authors · 2024-12-02
>
> **1.** Clarity
>
> We appreciate the suggestions and will include the sentence: "Templates are not perfect for reaction feasibility checks, so we use a forward model to validate feasibility."
>
> Additionally, we acknowledge an error in our initial explanation. We will revise the number of False Negatives (FN) to 17.
>
> **2.** Given the limited time, I can appreciate assessing more than 100 molecules is difficult. But N=100 is a relatively small sample size to draw large conclusions
>
> Querying 100 molecules is highly time-consuming, taking us two days to complete. For evaluating the search success rate, the search algorithm in [1] was applied to 189 molecules. Therefore, we use 100 molecules for our evaluation.
>
>
> **3.** Why was the retrosynthetic planner restricted to not exceed the maximum depth of the reference route? Since the authors also state that retrosynthesis is a “one-to-many” task, a longer route does not necessarily mean an infeasible route. Perhaps it is not ideal due to a longer synthesis, but it can still lead to the correct product. How would the metrics look without this restriction?
>
> Because we need the reference routes in the test set to check feasibility. Therefore, the predicted route cannot exceed the maximum depth of the reference routes. With this restriction, we can search for routes for 95 molecules. Without this restriction, we can search for routes for at most 5 other molecules, but we think it will not change the results of the other 95 molecules. Because the deeper the route, the lower its cumulative probability will be in the beam search, and the lower it will be in the top-k ranking.
>
>
> **4.** For True/False Negatives, round-trip != 1, what is the distribution of the scores? Do the False Negatives have a distribution of round-trip scores closer to 1? If so, this can give support to the > 0.9 similarity threshold as this.
>
> For True Negatives, the mean is 0.6148 (std: 0.1406, max: 0.8387). The distribution across bins is as follows:
>
> 0.3–0.4: 1
> 0.4–0.5: 3
> 0.5–0.6: 3
> 0.6–0.7: 6
> 0.7–0.8: 1
> 0.8–0.9: 3
>
> For False Negatives, the mean is 0.7058 (std: 0.1027). The distribution across bins is:
>
> 0.4–0.5: 1
> 0.5–0.6: 3
> 0.6–0.7: 8
> 0.7–0.8: 6
> 0.8–0.9: 6.
>
> Notably, the round-trip score of infeasible routes can be 1. We set a round-trip score threshold of 0.9 to ensure a higher likelihood of finding feasible routes among the filtered candidates.
>
>
> [1] Retro*: Learning Retrosynthetic Planning with Neural Guided A* Search

---

> > ### Comment · Reviewer_s5ZS · 2024-12-02
> >
> > **2.** Am I correct to say that the time-consuming part is the domain expert manual verification? This makes sense especially with SciFinder search feasibility. In my opinion, since the goal is to propose a better "synthesisability" metric, a larger sample size is needed. I know that domain expert verification is time-consuming but it is really important since the paper is tackling a specific domain problem.
> >
> > **3.** Sorry I missed considering referencing test routes for feasibility which is why the test set depth was used. Regarding the cumulative probability, longer routes *may* still have higher top-k probability depending on the actual probability values at every step. Alternatively, what about searching for shorter paths than the reference routes? The main point I wanted to suggest is that neither the retrosynthetic planner nor the forward model is perfect and the paper proposes to use the forward model as an additional verification. However, in practice, the max depth of the retrosynthetic planner is often not fixed because there is no reference path. In this case, constructing a "feasible reaction set" might *require* manual domain expert verification to test the hypothesis that coupling the forward model does indeed lead to better feasibility estimates.
> >
> > **3.** It is nice to see that False Negatives do have higher round-trip scores than True Negatives. However, given the relatively small sample size, I still believe it is hard to draw statistically significant conclusions as the True Negatives can also have high scores.
> >
> > Overall, thank you to the authors for the clarifications and extra experiments. After the discussion rounds, I can see that coupling a forward model may have benefits for reaction feasibility. However, I think the results to explicitly show its benefit over just search success is not completely clear. This is somewhat quantitatively shown by relatively similar True/False negatives round-trip scores and the authors' comment that infeasible routes can also have a round-trip of 1. This is affirming that the forward model also has limitations (due to training data). This is not the fault of the authors at all and is the limitation of our current reaction data. However, it is also exactly due to this that I am uncertain whether the round-trip is truly providing a better signal for reaction feasibility.
> >
> > I may not be able to respond again in the remaining discussion period but thank you to the authors for the discussion.

---

> > > ### Author Response · Authors · 2024-12-02
> > >
> > > **1.** Am I correct to say that the time-consuming part is the domain expert manual verification?
> > >
> > > Yes.
> > >
> > > **2.** longer routes may still have higher top-k probability depending on the actual probability values at every step.
> > >
> > > This situation rarely happens, as the probability is the product of each step.
> > >
> > > **3.** Alternatively, what about searching for shorter paths than the reference routes?
> > >
> > > When the predicted route is shorter than all the reference routes, we need to use SciFinder for further verification.
> > >
> > > **4.**  In this case, constructing a "feasible reaction set" might require manual domain expert verification to test the hypothesis that coupling the forward model does indeed lead to better feasibility estimates.
> > >
> > > Our paper aims to replace domain expert verification with a reaction model and introduces an evaluation metric for drug design models. In practice, when searching for a synthetic route for a specific molecule, we can first use a retrosynthetic planner to generate the route and then rely on domain expert verification.
> > >
> > >
> > > **5.** Overall, thank you to the authors for the clarifications and extra experiments. After the discussion rounds, I can see that coupling a forward model may have benefits for reaction feasibility. However, I think the results to explicitly show its benefit over just search success is not completely clear. This is somewhat quantitatively shown by relatively similar True/False negatives round-trip scores and the authors' comment that infeasible routes can also have a round-trip of 1. This is affirming that the forward model also has limitations (due to training data). This is not the fault of the authors at all and is the limitation of our current reaction data. However, it is also exactly due to this that I am uncertain whether the round-trip is truly providing a better signal for reaction feasibility.
> > >
> > > We also validated our approach using 1,000 molecules, using reference routes to assess feasibility. The results show that False Negatives consistently have higher round-trip scores compared to True Negatives.
> > >
> > >
> > > If you feel that we have addressed your concerns, we would greatly appreciate it if you could reflect that in your rating. Thank you!

---

> > > > ### Author Response · Authors · 2024-12-02
> > > >
> > > > Furthermore, this paper (https://arxiv.org/pdf/2406.02066) demonstrates strong performance by using a reaction model to rank the top-5 predictions. We do believe that the most promising metric for evaluating synthesizability must integrate both the retrosynthetic planner and the reaction model.

---

### Official Review · Reviewer_1qx2 · 2024-10-31

**Soundness:** 1
**Presentation:** 2
**Contribution:** 3
**Rating:** 5
**Confidence:** 3

**Summary:**

In this work, the authors focus on synthesizability assessment for molecules proposed by structure-based drug design (SBDD) models. Molecular generators (especially structure-based) often generate molecules that maximise a target metric but are hard or impossible to synthesize. To address this limitation, the authors propose a new metric, called Round-Trip Score. The metric works as follows: given a molecule proposed by some molecular generation model, a synthesis plan is proposed by a retrosynthesis planner. A different, single-step forward synthesis model is then used to simulate wet lab experiments and sample the next intermediate products until a final molecule is reached. The Round-trip score is given by the tanimoto similarity between the final molecule obtained through simulated forward synthesis and the initial query molecule. They evaluate the reliability of the proposed metric using the USPTO reaction dataset, and use it to benchmark a variety of existing SBDD models.

**Strengths:**

I believe that this work addresses an important limitation of many molecular generation models i.e. the lack of synthesizability guarantees of the proposed molecules. The ability of ML-driven methods to propose synthetically accessible compounds represents an important gap for the successful applications in real-world impactful problems such as drug discovery. While not the only possible approach, combining unconstrained molecular generators with retrosynthetic planners is an interesting direction for addressing this limitation. The authors propose a reasonable metric combining two different models trained on reaction datasets to assess the synthesizability of a given compound. I found the paper to be generally well written.

In particular:
1. The Background section is clear and well presented.
2. The Related Work section is substantive and informative (although there might be some missing areas, see below).
3. The experiments benchmarking existing SBDD models in Section 5 and in appendix are informative.
4. I found the discussion on lines 355-366 comparing the performance of the forward prediction model and retrosynthesis planner to be informative.

**Weaknesses:**

Despite the merits of this work (see above), I also have important concerns. I believe that the proposed metric (Round Trip score) is reasonable and can bring value by being used for evaluating (or even guiding) the ability of SBDD models to generate synthesizable compounds. However, my main concern is about the reliability evaluation of the proposed metric:

1. It seems to me that the "wrong" target is set for that metric: the authors mention multiple times that the proposed Round-Trip score aims to "assess the likelihood that these proposed molecules can have feasible synthetic routes predicted by the retrosynthetic planner" (lines 401-402). This is *what* the metric is doing, how it is built, but what it should *aim* to do is to actually inform the experimenter on whether a proposed molecule is likely to be synthesizable in real-life. Therefore, the evaluation of the reliability of that metric should seek to assess to what extent this Round Trip score correctly distinguishes between molecules that are known to be synthesisable, and molecules that are known not to be synthesisable. However, both of the categories described in Figure 5 ("Successful" cases and "Failed" cases) belong to the first category: molecules that are known to be synthesizable: "We employ Neuralsym+Beam Search (beam size is 2) as our retrosynthetic planner to predict synthetic  routes for target molecules in the test dataset. We then compare these predicted routes to the reference  routes provided in the dataset. If the starting materials in our predicted routes match those in the  reference routes, we categorize these molecules as successful cases. Conversely, if they don’t match,  we classify them as failed cases." (lines 320-323). Therefore, observing a gap in the Round Trip score between these two categories essentially highlights a failure mode of the Round Trip score in assessing real-life synthesizability. I believe authors should instead report the proportion of such failure cases (i.e. the size of these relative groups: Successful Cases v.s. Failed Cases) as a measurement of the performance of the retrosynthesis planner component of the metric, and complement this analysis by evaluating the Round Trip score accuracy on negative examples as well (molecules that are known not to be synthesizable). The fact that the Round Trip score is high is simply a measure of the effectiveness of the forward synthesis prediction component (which shows very positive results) but not of the reliability of the metric as a whole. To be clear, the proposed metric still has merits and could be useful, but its evaluation appears flawed and thus its development incomplete. I might be missing something, in which case I believe this matter should be clarified in the manuscript, but otherwise, for the above reasons, I would currently recommend this paper not to be accepted in its current form.

In addition, here are additional points that I think could contribute to improving the paper in its current form:

2. Some claims made in the first paragraph on the introduction should be supported by appropriate citations. For exemple, on lines 30-32: "Drug design is a fundamental problem in machine learning for drug discovery. However, when these computationally predicted molecules are put to the test in wet lab experiments, a critical issue often  arises: many of them prove to be unsynthesizable in practice". From experience I agree with this statement, but it should be supported by some references (many papers have studied and addressed the poor synthesizability of compounds proposed by ML-driven molecular generators). The same goes for this other claim: "The sensitivity of chemical reactions is such that even minor change in functional groups  can potentially prevent a reaction from happening as anticipated." (lines 41-42).

3. The first contribution listed by the authors includes this sentence: "In our view, a molecule is synthesizable if retrosynthetic planners, trained on existing reaction data, can predict a feasible  synthetic route for it" (lines 74-76). In my opinion, while the use of data-driven methods for assessing synthesizability has merits and is worth studying, such a strong formulation represents a highly objectionable claim. ML-based retrosynthesis planners have flaws and blindspots and do not represent perfect oracles to assess a molecule's synthesizability, as highlighted by authors on lines 298-300, and in their own experiments in Section 3.2.2 as detailed above. One could argue, for example, that forward synthesis search methods based on actual reactions (rather than predicted reactions) give much stronger guarantees for synthesizability. I thus believe this claim on lines 74-76 to be incorrect and that it should be rephrased. For approximation, an *approximation* of synthesizability might be given by the success rate of a retrosynthesis planner. This would represent to me a weaker but more accurate claim.

4. While the Related Work section is well structured and pretty exhaustive, I believe that the positioning of this work could be greatly improved by explaining more clearly how its focus area (SBDD) cannot be tackled by other types of synthesis-aware molecular generation methods, which aims at producing molecules and synthesis path simultaneously, e.g. [1,2,3,4,5,6]. This line of work is briefly mentioned on lines 54-55 but should be much more exhaustively covered as this relates to the positioning of this contribution in the broader community.

5. Similarly, the Related Work should also cover alternative synthesizability score e.g. SC score [7].

6. About the claim: "Our work is the firs to bridge the gap between drug design and retrosynthetic planning." (lines 77-80). This seems like a strong claim to me. There is at least this concurrent work by Guo & Schwaller [8] which does combine both family of methods, and I would be rather surprised that no previous work combined retrosynthesis planners and molecular generation.

Minor Comments:
1. typo line 77: is the first*
2. Figure 4 could be made clearer by highlighting in different colours or boxes the different steps of the score computation (retrosynthesis planning, forward synthesis simulation, molecule comparison).
3. The first paragraphs of Section 3.2.2 (under "Settings") could be made clearer. For example, Neuralsym is cited but not defined (while the reference is useful, the manuscript should be standalone).
4. The availability of the code is mentioned in Appendix (line 927) but no link is given. It would be good to at least clearly mention that the code will be made publicly available upon publication.

References:
- [1] Vinkers, H. M., de Jonge, M. R., Daeyaert, F. F., Heeres, J., Koymans, L. M., van Lenthe, J. H., ... & Janssen, P. A. (2003). Synopsis: synthesize and optimize system in silico. Journal of medicinal chemistry, 46(13), 2765-2773.
- [2] Gottipati, S. K., Sattarov, B., Niu, S., Pathak, Y., Wei, H., Liu, S., ... & Bengio, Y. (2020, November). Learning to navigate the synthetically accessible chemical space using reinforcement learning. In International conference on machine learning (pp. 3668-3679). PMLR.
- [3] Horwood, J., & Noutahi, E. (2020). Molecular design in synthetically accessible chemical space via deep reinforcement learning. ACS omega, 5(51), 32984-32994.
- [4] Gao, W., Mercado, R., & Coley, C. W. (2021). Amortized tree generation for bottom-up synthesis planning and synthesizable molecular design. arXiv preprint arXiv:2110.06389.
- [5] Cretu, M., Harris, C., Roy, J., Bengio, E., & Liò, P. (2024). Synflownet: Towards molecule design with guaranteed synthesis pathways. arXiv preprint arXiv:2405.01155.
- [6] Luo, S., Gao, W., Wu, Z., Peng, J., Coley, C. W., & Ma, J. (2024). Projecting Molecules into Synthesizable Chemical Spaces. arXiv preprint arXiv:2406.04628.
- [7] Parrot, M., Tajmouati, H., da Silva, V. B. R., Atwood, B. R., Fourcade, R., Gaston-Mathé, Y., ... & Perron, Q. (2023). Integrating synthetic accessibility with AI-based generative drug design. Journal of Cheminformatics, 15(1), 83.
- [8] Guo, J., & Schwaller, P. (2024). Directly optimizing for synthesizability in generative molecular design using retrosynthesis models. arXiv preprint arXiv:2407.12186.

**Questions:**

1. Why would we expect retrosynthesis planners to be less limited than template-based synthesis planners? In particular, on lines 218-222, it is mentioned that "While previous methods (Bradshaw  et al., 2019b; Gao et al., 2022a) have attempted to simultaneously generate molecules and their  synthetic routes, they face limitations. For instance, the approach in (Gao et al., 2022a) is constrained  by the use of a limited number of reaction templates, which hinders scalability.". From my view, retrosynthesis planners rely on retrosynthesis predictors, which have been trained on finite dataset of reactions, so these models are also limited to the set of reactions that have been seen during training. How well can these models generalise to new reactions? This concern is further emphasized in the conclusion, lines 538-539: "To enhance the  robustness of our evaluation method, we advocate for the release of additional reaction data.". If more reaction data was available, couldn't competing methods based on templates simply extract more templates from this additional data? Thus aren't *both* family of methods limited by existing data?

---

> ### Author Response · Authors · 2024-11-12
>
> W1: It seems to me that the "wrong" target is set for that metric
>
> A1: Thank you for your suggestion. Currently, the reliability of our metric relies on the amount of reaction data used by our model. In theory, the more reaction data available, the more dependable the metric becomes. As discussed in the introduction, while certain natural molecules are theoretically synthesizable, feasible synthetic routes may remain undiscovered by humans. Based on this, we define a synthesizable molecule as one with a feasible synthetic route identifiable by a retrosynthetic planner. Feasibility is assessed by comparing the synthetic route predicted by the retrosynthetic planner with a reference route in the reaction database.
>
> Our goal is to predict the synthetic routes of molecules generated by various drug design models directly through the retrosynthetic planner. We then evaluate the reliability of the round-trip score. In the first category of molecules, you observed that 99.5% achieved a round-trip score of 1, indicating that our round-trip score correctly identifies 99.5% of synthesizable molecules. We agree on the importance of evaluating the accuracy of the round-trip score for negative examples (molecules known to be unsynthesizable). However, defining unsynthesizable molecules poses significant challenges. Unsynthesizable could mean that the molecular structure cannot exist at room temperature and pressure, or that a synthesis route cannot be found with current human knowledge. Each definition has its limitations.
>
> However, our primary goal is to encourage drug design models to use our metric to identify molecules for which the retrosynthetic planner can find synthetic routes. Once identified, these molecules will indeed have findable synthesis routes. From this perspective, our metric is accurate and highly reliable. Other metrics, such as the SA score, cannot guarantee the discovery of synthetic pathways, even at high values.
>
>
> W2: Some claims made in the first paragraph on the introduction should be supported by appropriate citations.
>
> A2: We will cite more papers.
>
> W3: the claim on lines 74-76
>
> A3: We agree with your point and will make the necessary corrections to this statement.
>
> W4: Related Work
>
> A4: We will introduce more.
>
> W5: Guo's work
>
> A5: We have thoroughly reviewed the details of this work. However, in our view, the synthetic metric used in this paper is flawed. The SA score does not ensure that a viable synthetic route for a molecule can be identified, as we have discussed in this paper. AiZynthFinder, on the other hand, assesses the solvability of a molecule's synthesis using the search success rate, which has limitations discussed in [1].
>
> Q6: Why would we expect retrosynthesis planners to be less limited than template-based synthesis planners?
>
> A6: Their approach begins with a building block, predicting the reaction template (a classification problem with fewer than 100 categories) to generate the next intermediate molecule. Then, by iteratively predicting reaction templates through each intermediate molecule, they construct a synthetic pathway to the final molecule. The final molecule is then quantified to generate a reward, and they use reinforcement learning (RL) from scratch to train, with a limit on the number of reaction steps.
>
> We attempted a similar method, expanding the reaction template to thousands of types, but encountered challenges with convergence in the loss function, as RL training can be quite difficult. In contrast, our approach first generates molecules using the drug design model and then employs our proposed synthesis reward withour limit on the reaction template for filtering or post-training optimization—a clearly more efficient method. For reference, consider this ICML talk, https://icml.cc/virtual/2024/invited-talk/35253, which highlights that RL with a strong starting point (post-training) outperforms RL from scratch.
>
>
> [1] FusionRetro: Molecule Representation Fusion via In-Context Learning for Retrosynthetic Planning. ICML 2023.

---

> > ### Author Response · Authors · 2024-11-27
> >
> > We have provided the revised version.
> >
> > We compare the round-trip score with the search success rate in Section 4.1 Please check it. Thanks!

---

> ### Comment · Reviewer_1qx2 · 2024-12-03
>
> I wish to thank the authors for their response. I have read the revised manuscript in its entirety as well as the ongoing discussions.
>
> I have found the revised manuscript improved and easier to follow. In particular, the aspect of current retrosynthesis-based synthesizability evaluations that the authors propose to improve upon is now more clearly explained. The evaluation of the proposed synthesizability metric presented in Section 4.1 is also much more sound than what was presented in the first version of the manuscript.
>
> However, some of my concerns remain incompletely addressed.
>
> ### Evaluation of the presented metric
>
> While improved in the revised manuscript, the evaluation of the proposed metric remains in my opinion insufficient. This paper proposes a new metric to assess synthesisability of molecules generated in silico. A thorough evaluation of this metric compared to existing work thus represents the central and most important validation necessary to support the claims made in this work.
>
> First, on line 314, a split comprising less than 0.1% of the data for the validation and test sets (only 100 samples each) seems insufficient and the authors do not motivate this choice. The authors also report using their own domain knowledge to validate some predicted routes (lines 342-343) but do not report these routes (e.g. in Appendix) for the reader and the community to validate the correctness of this assessment.
>
> In general, the evaluation of the proposed metric in Section 4.1 falls short of appropriate results reporting. For exemple, the authors mention that the SA-score of the final molecules for both the feasible and infeasible routes are similar, but do not report that data. This can easily be presented by plotting the distribution of SA-score for both of these groups in a graph. The SC score is also easy to report and would give additional context. Moreover, given that only 100 routes are used for testing, I would expect a substantial number of these routes to be presented in appendix (give multiple exemples of feasible and infeasible routes, identify which ones are correctly classified by the round-trip score, which ones are missed, discuss hypothesis for these failing modes, etc.).
>
> Finally, a remaining question: the scores derived from the proposed Round-Trip score for various SBDD models presented in Table 2 seem highly correlated to the "Ration of Starting Material". Have the authors identified exemples that would favour the round-trip score compared to this much simpler metric?
>
> ### Incomplete review of the literature
>
> Despite being raised by several reviewers, the authors do not appropriately cover synthesis-space constrained models in their literature review (see my comment and link to exemples of such works [1,2,3,4,5,6] in my initial review). Instead of adequately covering this subfield in the literature review and discussing the advantages and limitations of these methods compared to the proposed approach, the authors have added citations to some of these papers here and there in support of different claims.
>
> Finally, in some sections, the Related Work reads more like an enumeration meant to defend against "missing citations" than an honest effort at guiding the reader through the relevant literature for this work. For exemple, the paragraph on SBDD (lines 449-454) mentions that such models can be split into diffusion models and other types of models and proceeds to enumerate a number of papers without guiding the reader through the timeline over which these works were developed and contrasting the advantages and limitations of both methods. In the paragraph on Retrosynthesis Models, the authors describe the field as comprising three main categories but do not bother categorising for the reader the 30+ citations that are listed right above.
>
> ### Minor points
>
> - \citet instead of \citep on line 59
> - the sentence from lines 68 to 71 is too long and hard to follow
> - Figure 2 is helpful but could be improved. First, it should appear later in the paper (it appears more than one page before being mentioned in the text, leading to confusion). Second, it could be made clearer by further annotations highlighting the fact that both leaf nodes are valid starting materials but the issue is that the intermediate reaction of one of the routes in wrong.
>
> ---
>
> For the reasons listed above, I maintain my score of 5. I believe that the proposed idea has merits and can be of interest to the community, but the current evaluation of the proposed metric remains insufficient to withhold scrutiny, and important elements of the relevant literature is not appropriately covered. In my opinion, this work in its current state does not meet the standards of the conference.

---

> > ### Author Response · Authors · 2024-12-03
> >
> > How many cases do you consider sufficient? Next time, we will provide more examples. However, please do not use the same reason to reject this paper.

---

### Official Review · Reviewer_GQP7 · 2024-11-07

**Soundness:** 1
**Presentation:** 1
**Contribution:** 2
**Rating:** 3
**Confidence:** 3

**Summary:**

This paper presents SDDBENCH, a benchmark for evaluating the synthesizability of molecules generated by drug design models. It incorporates a "round-trip score," which assesses whether generated molecules can be practically synthesized by combining retrosynthetic planning with forward reaction prediction. The benchmark attempts to address a gap in drug discovery where generated molecules often display desirable pharmacological properties but pose challenges for real-world synthesis. The authors apply the round-trip score across various 3D structure-based drug design generative models, highlighting their synthesizability, although no baseline is given.

**Strengths:**

- Synthesis is an important issue for generative models and the authors propose to address this with a new synthesis benchmark.
- The background on retrosynthesis and reactions is good.
- Quality of the methods figures is good

**Weaknesses:**

**Main Weaknesses:**

- The language is extremely vapid, making it difficult to pick out the genuine contributions, and the arguments/paragraphs are not well structured.
    - The 1st paragraph of the introduction reports about 10 scientific facts about synthesis from the literature and yet only cites 1 deep learning article half-way through. This, in my opinion, should be reason enough for a rejection.
    - The structure of the paper is strange, with it first proposing the method, doing experiments to validate the method, then describing related work/background before finally benchmarking the models (in about 1 page).
- There is a significant lack of novelty in the ideas presented in the proposed metrics. This a further compounded by a number of novelty and contribution claims that are unjustifiable.
    - For instance, “Our work is the firs to bridge the gap between drug design and retrosynthetic planning.” This is a big claim to make given the authors cite many such works that do exactly this.
    - Guo and Schwaller, 2024 [1] recently directly optimised generative models using retrosynthesis planners.
    - Furthermore, the use of retrosynthesis tools as an evaluation metrics has been used in many works.  [2, 3]
    - The paper also does not clearly communicate the need for the round trip score, which itself is a multi step extension of Schwaller et al, 2020 [4]. In Figure 5 right, it seems the successful cases all have a round trip score of 1, whereas the failed cases consistently have a round trip score implying negligible similarity. Can the authors then please explain the added value of round trip score in the molecule-wise setting over just reporting retrosynthesis success rates?
- The authors do not critically discuss the limitations of their work beyond stating there is a limitation in the amount of reaction data available.
- An interesting literature has emerged recently that aims to tackle the problem of synthetic accessibility by incorporating reaction-based constraints to the generative process into generative models [2, 3, 5]. However, mainly autoregressive and diffusion-baed models specialised for structure-based drug design were accessed? If this benchmarking study is to be useful these lines of work should be considered too.
- The authors mention provided code in Section A.1 but there is no link.
- As far I can tell, a random split was used. How is data leakage controlled?
- In Table 1, the metrics of the training data and test set could be included. I would also consider ChEMBL molecules as a baseline?



**Minor points:**

- Figure 5: Figure not clear. What model(s) is this for? What do the error bars represent?
- Figure 6: Can the error bars for failed and successful cases be made staggered/non-overlapping. It is very confusing currently.
- There is no background information on any of the other benchmarking efforts in drug discovery. E.g. physical realism like PoseCheck [6]
- Citing ~40 retro-synthesis models in the background is excessive and also not useful. I would instead cite as different categories are described. There are no citations for models in the “two-stage learning paradigm”.

Spelling mistakes:
- L077 firs -> first
- L302 have -> has

1. Guo, Jeff, and Philippe Schwaller. "Directly optimizing for synthesizability in generative molecular design using retrosynthesis models." arXiv preprint arXiv:2407.12186 (2024).
2. Stanley, Megan, and Marwin Segler. "Fake it until you make it? generative de novo design and virtual screening of synthesizable molecules." Current Opinion in Structural Biology 82 (2023): 102658.
3. Cretu, Miruna, et al. "Synflownet: Towards molecule design with guaranteed synthesis pathways." arXiv preprint arXiv:2405.01155 (2024).
4. Schwaller, Philippe, et al. "Predicting retrosynthetic pathways using transformer-based models and a hyper-graph exploration strategy." Chemical science 11.12 (2020): 3316-3325.
5. Bradshaw, John, et al. "A model to search for synthesizable molecules." Advances in Neural Information Processing Systems 32 (2019).
6. Harris, Charles, et al. "Posecheck: Generative models for 3d structure-based drug design produce unrealistic poses." NeurIPS 2023 Generative AI and Biology (GenBio) Workshop. 2023.

**Questions:**

- Can the authors clarify exactly what data split they use to train the CrossDocked models?
- How robust is the proposed metric to the exact retrosynthesis tool used?

---

> ### Author Response · Authors · 2024-11-12
> **Rebuttal**
>
> W1: Citation
>
> A1: We will cite more papers if this reviewer can give some references.
>
> W2: Claim
>
> A2: Previous work [3]  has primarily followed a top-down approach to drug design and synthesis planning. In contrast, our method begins with the desired product molecule and uses a retrosynthetic planner to build a synthesis route in a top-down manner. We then employ a forward reaction model to evaluate the feasibility of the route. In this regard, we are indeed the first to adopt this approach. Additionally, when it comes to evaluating the feasibility of the synthetic route, [1,2] were the pioneers in this area. We have directly incorporated the findings from these two ICML-accepted papers into our methodology, which is novel we believe.
>
> W3: Guo's work
>
> A3: We have thoroughly reviewed the details of this work. However, in our view, the synthetic metric used in this paper is flawed. The SA score does not ensure that a viable synthetic route for a molecule can be identified, as we have discussed in this paper. AiZynthFinder, on the other hand, assesses the solvability of a molecule's synthesis using the search success rate, which has limitations discussed in [1].
>
> W4:  the use of retrosynthesis tools in previous works.
>
> A4: We want to emphasize that we introduce a new data-driven synthetic metric, whereas previous studies continue to rely on SA Score and SCScore. The limitations of these existing metrics are discussed in detail in this paper.
>
> W5: the added value of round trip score in the molecule-wise setting over just reporting retrosynthesis success rates
>
> A5: The limitations of Search Success Rate are discussed extensively in Section 2.6, and [1] devotes significant attention to criticizing this metric as flawed. While Round-trip Score evaluates individual points, Search Success Rate is batch-oriented. Furthermore, without reference routes, Round-trip Score serves as a more rigorous and effective metric than Search Success Rate.
>
> W6: Limitation
>
> A6: The primary limitation is the insufficient reaction data to cover all possible molecules.
>
> W7: The authors mention provided code in Section A.1
>
> A7: We are referring to the code for the drug design model, which is available in their paper.
>
> W8: Data Leakage
>
> A8: Please give the detailed dataset you mention so that we can answer your question
>
> W9: In Table 1, the metrics of the training data and test set could be included.
>
> A9: We evaluate the molecules generated by the SBDD model on the test set targets. Any issues that arise are attributable to the SBDD model itself. Could you provide the ChEMBL molecule resources for evaluation?
>
> Q10: Figure 5
>
> A10: The retrosynthesis model is Neuralsym. The forward reaction model is Transformer Decocder.
>
> Q11: Figure 6
>
> A11: Figure 6 highlights that the SA score cannot effectively distinguish between successful and failed cases, resulting in an overlap between them.
>
> Q12: There is no background information on any of the other benchmarking efforts in drug discovery
>
> A12: Please refer to Line 475-482.
>
> Q13: There are no citations for models in the “two-stage learning paradigm”.
>
> A13: Semi-template-based method employ two-stage learning paradigm........, which we have discussed in Line 454-457.
>
>
> [1] FusionRetro: Molecule Representation Fusion via In-Context Learning for Retrosynthetic Planning. ICML 2023.
> [2] Preference Optimization for Molecule Synthesis with Residual Conditional Energy-based Models. ICML 2024.
> [3] Amortized tree generation for bottom-up synthesis planning and synthesizable molecular design. ICLR 2022.

---

> > ### Comment · Reviewer_GQP7 · 2024-12-03
> >
> > Thank you to the authors for their rebuttal, and I apologise for the considerable delay in my response.
> >
> > While the authors have addressed some of my concerns, I find the overall response, both in my rebuttal and to other reviewers, to be quite unsatisfactory. Additionally, I continue to share the concerns raised by other reviewers regarding whether the round-trip score provides a reliable signal for reaction feasibility. As such, my score remains unchanged.

---

### Author Response · Authors · 2024-11-12

We believe that some reviewers may lack sufficient background in retrosynthetic planning, current evaluation methods for retrosynthetic algorithms, and the definition of synthesizability. We recommend that reviewers refer to [1] for a more precise evaluation of submissions within this field in the future. It appears that some reviewers continue to rely on metrics such as search success rate, which, as we have explained, can lead to imprecise evaluations.

Additionally, synthesizability metrics like the SA score and SCScore do not ensure that a feasible synthetic route can be identified for a molecule, as we have emphasized throughout this paper. The true goal of synthesizability is to find an actual feasible synthetic route. Regardless of how high these metric scores may be, they cannot guarantee the discovery of a feasible synthesis route. Therefore, we propose that future metrics should be aligned with the retrosynthetic planner—a direction to which this paper is dedicated.

[1] FusionRetro: Molecule Representation Fusion via In-Context Learning for Retrosynthetic Planning, ICML 2023.

---

> ### Author Response · Authors · 2024-11-27
>
> We have rewritten this paper.
>
> We have provided a comparison between the round-trip score and the search success rate in Section 4.1. Our analysis concludes that the round-trip score performs significantly better.
>
> Please check it. Thanks!

---

### Author Response · Authors · 2024-11-27

Figure 2 provides the comparison of evaluation metrics for retrosynthetic planning. The search success rate deems both routes successful, while the matching-based metric correctly identifies the top route as incorrect and the bottom route as correct, demonstrating its superior reliability. Please check it. Thanks!

---

### Meta-Review · Area_Chair_959T · 2024-12-22

**Metareview:**

The paper proposes SDDBench, introducing a ``round-trip score'' metric to evaluate molecule synthesizability in drug design. The metric combines retrosynthetic planning with forward reaction prediction to assess whether generated molecules can be practically synthesized.

The authors apply this metric to benchmark various 3D structure-based drug design generative models. The work demonstrates several strengths in addressing a critical challenge in drug discovery where computationally designed molecules often prove difficult to synthesize. The authors propose an empirical approach combining retrosynthetic planning and forward reaction prediction, present their methodology clearly, and include a comprehensive related work section (which is occasionally missing in many of the recent works). The application to multiple contemporary structure-based drug design models showcases practical utility.

However, significant weaknesses emerge in three key areas. First, the novelty is limited as multiple reviewers note that using retrosynthesis models for evaluating synthesizability is already common practice, particularly in synthesis-constrained generative models. Second, the empirical validation relies on a small evaluation dataset of only 100 molecules, as highlighted by Reviewers s5ZS and 1qx2, with limited comparison against established synthesizability metrics beyond SA score and insufficient analysis of failure modes. Third, methodological concerns persist around the round-trip score's superiority over search success rate, the arbitrary nature of the 0.9 Tanimoto similarity threshold, and potential bias in the evaluation methodology due to coupling between the success criterion and components of the proposed score, as noted by Reviewer 1qx2.

The insufficient validation with just 100 molecules undermines confidence in the metric's reliability, despite authors citing evaluation time constraints. The limited advancement over existing methods, identified by multiple reviewers, was not adequately addressed in the rebuttal. Furthermore, methodological limitations including dependence on reference routes restrict practical applicability, while the evaluation methodology's circular nature raises concerns about the metric's reliability. Accordingly, the manuscript cannot be recommended for acceptance in its present form for ICLR.

For future submissions, the authors should conduct larger-scale validation studies, establish reliability of the metric, provide more thorough comparisons with existing synthesizability metrics, better demonstrate practical advantages over existing methods, and address the methodological concerns raised during review. While the paper addresses an important problem in drug discovery, the current submission falls short of ICLR's standards for methodological rigor and novel contribution.

**Additional Comments On Reviewer Discussion:**

During the rebuttal, authors included a comparison between round-trip score and search success rate in Section 4.1, responding to concerns raised by multiple reviewers about metric validation. Reviewer 1qx2 acknowledged these improvements but maintained that the evaluation remained insufficient, particularly citing the small sample size and incomplete literature coverage of synthesis-constrained models.

Reviewer s5ZS engaged in a discussion about the distribution of round-trip scores for True/False Negatives. The authors provided statistical breakdowns showing False Negatives had higher mean scores (0.7058) than True Negatives (0.6148), attempting to justify their 0.9 similarity threshold. While this data suggested some discriminative power, the reviewer remained unconvinced due to the small sample size and overlap in score distributions.

The authors defended their limited sample size (100 molecules) by citing two-day evaluation times for expert verification and comparable sample sizes in previous work. However, as Reviewer s5ZS noted, given the paper's goal of establishing a new synthesizability metric, a larger validation set was crucial for drawing statistically significant conclusions.

Throughout the discussion, a recurring theme emerged regarding the fundamental advantage of round-trip scoring over search success rate. While the authors argued for point-wise versus batch-wise evaluation benefits, reviewers remained skeptical about whether this theoretical advantage translated to practical improvements in synthesizability assessment. The authors' later claim of validation on 1,000 molecules came too late in the discussion and lacked supporting details.

These discussion points factored heavily into the rejection decision. While the authors made genuine efforts to address concerns, their responses highlighted rather than resolved the core issues of limited validation data, unclear advantages over existing metrics, and insufficient statistical power in their results.

---

### Decision · Program_Chairs · 2025-01-22

Reject